# Various Biomimetics, Including Peptides as Antifungals

**DOI:** 10.3390/biomimetics8070513

**Published:** 2023-10-28

**Authors:** Elena Efremenko, Aysel Aslanli, Nikolay Stepanov, Olga Senko, Olga Maslova

**Affiliations:** Faculty of Chemistry, Lomonosov Moscow State University, Lenin Hills 1/3, Moscow 119991, Russia

**Keywords:** antifungal peptides, enzyme-like activity, growth inhibition, antifungal activity, fungal metabolites, quorum response molecules, combined antifungals

## Abstract

Biomimetics, which are similar to natural compounds that play an important role in the metabolism, manifestation of functional activity and reproduction of various fungi, have a pronounced attraction in the current search for new effective antifungals. Actual trends in the development of this area of research indicate that unnatural amino acids can be used as such biomimetics, including those containing halogen atoms; compounds similar to nitrogenous bases embedded in the nucleic acids synthesized by fungi; peptides imitating fungal analogs; molecules similar to natural substrates of numerous fungal enzymes and quorum-sensing signaling molecules of fungi and yeast, etc. Most parts of this review are devoted to the analysis of semi-synthetic and synthetic antifungal peptides and their targets of action. This review is aimed at combining and systematizing the current scientific information accumulating in this area of research, developing various antifungals with an assessment of the effectiveness of the created biomimetics and the possibility of combining them with other antimicrobial substances to reduce cell resistance and improve antifungal effects.

## 1. Introduction

Currently, there is an urgent problem of efficient prevention of plant, animal and human diseases triggered by pathogenic fungi [1,2,3]. Infections that cause plant diseases affect food crops annually and destroy the harvested crop during storage, which entails the formation of serious social and economic problems. In addition, among plant protection products, there are many fungicides which are toxic to humans, as well as those that have a negative impact on agrobiocenoses, causing environmental pollution and the accumulation of remaining amounts of pesticides in agricultural products [4]. Filamentous fungi (*Aspergillus*, *Cladosporium*, *Penicillium*, *Stachybotrys, Fusarium*) synthesizing various mycotoxins are particularly dangerous, both at the stage of crop growth and at the stage of storage [5].

With fungal (yeast) contaminations of humans and animals, special difficulties arise in the treatment of invasive infections characterized by high mortality [6]. Currently, various antifungals are used to combat such fungal pathogens as polyene antibiotics (nistatin, amphotericin B), azoles (fluconazole), modified lipopeptides (echinocandins), allylamines (terbinafine), etc. [7,8]. Most of them are synthetic compounds, and therefore, there are concerns about their high toxicity due to limited water solubility and the need to use their relatively high concentrations, as well as about the decrease in the effectiveness of their action due to the emerging resistance in fungi [9].

Interest exists in the natural compounds possessing strong antifungal activity with a low enough toxicity to other eukaryotic cells; however, most of such compounds appeared to be biodegradable by fungal strains. The fungal pathogens can use almost any natural substrates due to their ability to synthesize various hydrolases and oxidoreductases with a wide substrate specificity of action [10,11]. Having a unique ability to adapt to changing environmental conditions, fungi can overcome the effects of various antifungal agents. In addition, the frequent and irregular use of fungicides has led to the emergence of resistant strains [12]. At the same time, such resistance of fungi at the present stage of research of these microorganisms is also associated with the presence of a Quorum sensing (QS) mechanism similar to bacterial populations [13].

In this regard, the development of new antifungal agents is an urgent task, and there is a huge interest in biomimetics, which are substances imitating their natural analogs. Thus, natural antifungal peptides (AFPs) have demonstrated unprecedented advantages as selective biomaterials for obtaining effective biologics [14]. Therefore, AFPs obtained from natural sources (plants, fungi, bacteria), which can be chemically modified or artificially synthesized to improve their antifungal properties and reduce side effects, are of great interest [15]. When obtaining AFPs as biomimetics, the possibility of increasing their stability and bioavailability turns out to be very attractive.

In addition to AFPs, the synthesis of a number of other compounds and materials forming various surfaces imitating natural analogs that can be attributed to biomimetics is also interesting from a scientific and practical point of view. The antifungal effect of these compounds and materials is not limited to the destruction of the cell wall and membranes of fungi but affects their various metabolically important pathways, growth and spore formation [16,17,18]. Of great interest is the possible synergistic effect among such biomimetics with antifungal drugs, which allows for a reduction in the doses of individual substances used [19,20].

Thus, the purpose of this review was to analyze and summarize the main current trends in research on the production of antifungal biomimetics using various approaches (Figure 1) based on damage to the integrity of cells and individual organelles, disruption of synthetic processes in fungal cells, inhibition of the activity of various enzymatic systems, etc.

This review contains information about antifungal substances of both peptide and non-peptide nature with action aimed at different targets and with a wide range of possible applications, which makes it unique. A separate emphasis in this review was made on comparing the antifungal effect that various variants of biomimetics have on different fungal cells and on the possibility of combining these compounds (including enzymes that act on various fungal targets and metals that generate reactive oxygen species (ROS)) in order to identify the most effective and promising ones for further improvement and application.

## 2. Antifungal Peptides (AFPs)

Antimicrobial peptides (AMPs) produced by living organisms attract particular interest as new antimicrobial agents with their wide range of biotargets (bacteria, fungi, parasites) capable of replacing blockbuster antibiotics [21]. To date, there are a number of databases for documenting AMPs [22]. In this review, we considered information from four of them: the Antimicrobial Peptides Database (APD3), the Collection of Antimicrobial Peptides (CAMP_R3_), the Database of Antimicrobial Activity and Structure of Peptides (DBAASP), and the Database of Antimicrobial Peptides (dbAMPs) [23,24,25,26] (Figure 2).

APD3 is a system dedicated to the discovery timeline, glossary, nomenclature, classification, structure, information search, prediction, design, statistics and tools of AMPs and beyond. As of August 2023, the APD3 contains a total of 3569 AMPs. 

The dbAMP database, created by Dr. Lee’s team, contains information about 28,709 different types of AMPs in 3044 organisms, including experimentally verified AMPs. The information from such protein databases as UniProt, NCBI and Protein Data Bank was used for the construction of dbAMP.

The DBAASP database contains 20,876 AMPs and information about their chemical structure, amino acid sequences, target species and object, and hemolytic and cytotoxic activities of peptides. In addition, DBAASP provides a tool for the in silico prediction/design of new AMPs.

CAMP_R3_ contains information on the AMP sequences, protein definition, accession numbers, activity, source organism, target organisms, protein family descriptions and N and C terminal modifications. Currently, the database contains 8164 peptide sequences and also provides tools for sequence alignment, pattern creation and pattern and HMM-based searches.

The peptides in these databases can be classified into four categories: (i) natural AMPs; (ii) predicted peptides, which are predicted by machine learning or other technologies and tested to be active; (iii) synthetic peptides, derived from natural AMPs; and (iv) patented AMPs. Thus, the described AMP databases are an effective tool for the analysis, prediction and design of new peptides with desired properties, particularly AFPs. 

As of August 2023, the proportion of AFPs among all AMPs reported in the APD3, CAMP_R3_, DBAASP and dbAMPs databases was 36.7%, 14%, 28.6% and 19.3%, respectively (Figure 2). 

Interestingly, the ratio of synthetic AFPs (SAFPs) to natural AFPs varied depending on the database under consideration and was the maximum (68.1%) in the DBAASP database. 

Thus, it is obvious that covering, in one article, the huge number of peptides known is a nearly impossible task. When selecting substances for discussion in this review, we were guided by the number of studies conducted with each of the peptides, the presence of a known structure, and the presence of synthetic analogs in the case of the natural peptides under consideration. At the same time, the most recent and relevant synthetic peptides obtained by prediction and studied against various pathogens were also considered.

According to the data of Figure 2 and Table 1 [27,28,29,30,31,32,33,34,35,36,37,38,39,40,41,42,43]), a number of AMPs with antifungal activity have already been identified, which can be obtained from various natural sources. Due to the possibility of obtaining SAFPs capable of imitating various natural peptides, such biomimetics can find effective application in practice and are notable objects for current developments and investigations (Table 2 [29,34,39,44,45,46,47,48,49,50,51,52,53,54,55,56,57,58,59,60,61,62,63]).

It should be noted that AFPs can be classified by their total charge, secondary structure, mechanism of action, etc. In this review, when discussing AFPs, the main attention was paid to their origin (natural, semi-synthetic or synthetic), the object of their influence, the mechanism of their influence on fungal cells and the level of antifungal effect. Such information was summarized in Table 1 and Table 2.

Despite the fact that the mechanism of action of AMPs in relation to fungal cells is not as well studied as in relation to bacteria, the basic principle of their antifungal action is similar to the antibacterial one and most often consists in violation of the functions and integrity of the cell wall and plasma membrane. Due to this principle of action of AMPs, the development of resistance to them in microorganisms is considered to be almost impossible. AFPs, in addition to the mentioned effect on fungal cells, are able to specifically inhibit membrane proteins, β-1,3-glucan and chitin synthases, thereby contributing to the formation of defects in the cell wall, or inhibit H+-ATPase, causing an apoptosis-like process. Some AFPs can also induce intracellular generation of ROS, destroying various biomolecules (lipids, proteins, nucleic acids, etc.) by their active oxidation [64,65,66,67] (Table 1).

Semi-synthetic AFPs are chemically modified natural peptides while preserving the active centers of the origin molecule in order to achieve optimal properties [68,69]. Echinocandins (anidulafungin (#20), caspofungin (#21), micafungin (#22)), having a lipopeptide nature, are members of the “youngest” clinically used group of semi-synthetic AFPs. Echinocandins consist of a cyclic hexapeptide nucleus with a lipid side chain. The peptide lipid tail anchors the lipopeptide in the cell membrane next to the target enzyme. These peptides act as non-competitive inhibitors of a key enzyme in the synthesis of β-1,3-glucan [69]. Unlike semi-synthetic AFPs, SAFPs are obtained entirely by chemical synthesis. Most often, a solid-phase method is used for the synthesis of SAFPs, based on the addition of one amino acid at one step of the synthesis, which allows investigators to study the role of each amino acid in the synthesized sequence.

In addition to the methods of chemical modification and synthesis of AMP molecules, the use of methods of computer molecular design played a significant role in obtaining synthetic AMPs with the desired characteristics. This made it possible to combine information about the chemical properties and biological activity acquired by new peptides and the amino acid sequences present in them. This, in turn, made it possible to develop methods for predicting and evaluating the antifungal potential of synthesized sequences in silico [70].

### Examples of Synthetic Analogs of the Natural AFPs

When analyzing examples of currently created semi-synthetic and synthetic AFPs, the result of studying their properties and mechanisms of action, as well as comparing them with natural analogs, is always interesting and useful. Therefore, the main idea of the analysis was to show how appropriate it is to obtain certain synthetic analogs of natural peptides from the point of view of influencing the peptide properties (antimicrobial efficiency, toxicity, stability, mechanism of action, etc.). It appears that the targets of action of AFPs can be different, and this is due to their different chemical structures.

An important role in studying the antimicrobial properties of protein/peptide molecules is played by the study of their three-dimensional structures. The study of the active sites of protein/peptide molecules and mechanisms of action is an important tool in the search for effective therapeutic compounds. However, it should be taken into account that, unlike proteins, peptides can take on multiple conformations, and their 3D structures can change due to intermolecular interactions with solvent molecules and biomolecules (Figure 3) [71].

Antimicrobial peptides polymyxin B (#2) and colistin (polymyxin E) (#3) exhibited antifungal activity against 11 MDR yeast and filamentous fungal strains, including strains belonging to the *Candida, Cryptococcus* and *Rhodotorula* yeast genera, along with others belonging to *Aspergillus*, *Fusarium*, *Scedosporium*, *Lichtheimia* and *Rhizopus*, with MICs ranging from 16 to 128 μg/mL, except for the *Aspergillus* species [28].

It was shown that peptides naturally derived from milk protein lactoferrin, such as lactoferricin (Lfcin) (#5,6), lactoferrampin (Lfampin) (#4) and Lf(1-11) (#24) and some modified or synthesized peptides (#25) could also exhibit strong antifungal activity. In particular, the antifungal activity of bovine lactoferrin-derived Lfcin (#5) was investigated against a wide range of fungal species, and it was shown that bLfcin (#5) demonstrates strongly enhanced antimicrobial activity compared to lactoferrin [29,39,72,73,74].

Using a simple and reliable method, a set of anti-*Candida* peptide CGA-N12 (#23) analogs was rationally designed, and seven CGA-N12 analogs with significantly improved antifungal activity against *C. tropicalis* were screened [47].

An investigation of the effect of four antimicrobial peptides—PPD1 (FRLHF) (#26), 66-10 (FRLKFH) (#27), 77-3 (FRLKFHF) (#28) and D4E1 (FKLRAKIKVRLRAKIKL) (#29)—on the aflatoxin production by *A. flavus* and *A. parasiticus* suggested that AMPs at the near minimum inhibitory concentrations (MIC) were effectively inhibiting aflatoxins without hindering the growth of the fungi. At higher concentrations, these peptides exerted fungicidal action on *A. flavus* [48,49].

A C14-residue peptide named KK14 (#30), with the sequence KKFFRAWWAPRFLK-NH_2_, was designed and inhibited the conidial germination and fungal growth of food contaminants. The substitution of a Pro residue with Arg increased the helical content of the peptide, not only with its antifungal activity but also its cytotoxicity. The insertion of an unnatural bulky residue, β-diphenylalanine, or a full denantiomerization, overall increased the antifungal potency [50].

The small antimicrobial peptide PAF26 (Ac-RKKWFW-NH_2_) (#31) demonstrated multiple detrimental effects on the filamentous fungi *Penicillium digitatum*, which ultimately resulted in the permeation and killing of the growing cells [51].

Ultrashort peptide H-Orn-Orn-Trp-Trp-NH_2_ (O3TR) (#32) inhibited the growth of the filamentous fungi *Fusarium culmorum*, *Penicillium expansum* and *A. niger*, and the yeasts Saccharomyces cerevisiae, *Zygosaccharomyces bailii*, *Z. rouxii*, *Debaryomyces hansenii* and *Kluyveromyces lactis*. The addition of a C12 fatty-acid chain tail at the N-terminus of these peptides improved its antifungal activity by 2–8-fold in relation to different fungi [52].

The rationally designed and synthesized new structural class of dipeptides Trp-His(1-Bn)-OMe/NHBn and tripeptides His(1-Bn)-Trp-His(1-Bn)-OMe/NHBn, particularly Trp-His[1-(3,5-di-tert-butylbenzyl)]-NHBn (#33), possessing modified amphiphilic histidine along with hydrophobic tryptophan residues, demonstrated promising antifungal activity with membrane lytic action against *C. neoformans* [53].

The mechanism of action and activity of a series of synthetic analogs (#34) of the halictine HAL-2 peptide (#7) (from the venom of the wild bee *Halictus sexcinctus*) was investigated in relation to cells of *Candida* spp. It was found that halictines can rapidly permeabilize cell membranes and cause the leakage of cytosolic components and that their mode of action is likely to depend on the plasma-membrane sterols. Pre-treatment with the inhibitors of sterol synthesis (terbinafine and fluconazole) resulted in a significant reduction in peptide efficacy, while their killing efficacy increased when combined with amphotericin B [30,54].

A synthetic analog of antimicrobial peptide halocidin (#8), di-K19Hc (#35), has exhibited improved antifungal activity against a panel of fungi, including several strains of *Aspergillus* and *Candida* [31,55].

MSI-78 (pexiganan) (#36) is a synthetic peptide derived from naturally occurring magainin-2 (#9) produced by *Xenopus laevis*. An in vitro evaluation of the antifungal activity of MSI-78 against clinical isolates of *F. solani* demonstrated that MICs of this peptide can vary between 10 and 80 mg/L [32,33,39].

An investigation of the antifungal activity of *Ixodes ricinus* defensins (DefMT3, DefMT6 and DefMT7) (#10) and their γ-core motifs (#37) against *Fusarium* species demonstrated that the antifungal activity of the γ-core of selected peptides, particularly DefMT3 (#37), was higher than the full peptides [34].

It has been shown that a number of AFPs can exhibit not only antifungal activity but also inhibit the production of mycotoxins synthesized by fungi as molecules for their self-defense [48,64]. It turned out that, in some cases, individual AFPs may lose their ability to inhibit the growth of fungi but retain their effectiveness in inhibiting the biosynthesis of mycotoxins by fungi. For instance, the treatment of *F. graminearum* cells with the reduced form of the γ-core of the tick defensin DefMT3 (TickCore3, TC3) decreases the growth of the fungal cells and abrogates mycotoxin (trichothecene B) production. The oxidation of TC3 leads to the loss of its growth-inhibitory activity, while the anti-mycotoxin activity is retained [75].

The full-length *Neosartorya (Aspergillus) fischeri* AMPs (#11) and novel rationally designed γ-core peptide derivatives γ^NFAP^-opt and γ^NFAP^-optGZ (#38) exhibited high efficacy by inhibiting the growth of the agriculturally relevant filamentous ascomycetes in vitro [35].

Using an easy step-by-step way to choose, characterize and test potential sequences to be assayed as synthetic AMPs, two peptides (PepGAT (#39) and PepKAA (#40)) with antimicrobial potential against *Candida* spp., including activity against biofilms and without any hemolytic effects, were identified and characterized [57]. SAFPs PepGAT (#39) and PepKAA (#40) demonstrated strong inhibition of *P. digitatum* growth. All peptides targeted the fungal membrane, leading to pore formation, loss of internal content and death. The induction of high levels of ROS was also a mechanism employed by some peptides [56,57,58].

A small peptide, RcAlb-PepII (#41), designed based on the primary structure of Rc-2S-Alb, a 2S albumin from the seed cake of *Ricinus communis*, strongly inhibited the growth of *Klebsiella pneumoniae* and *Candida parapsilosis*, and induced morphological alterations in their cell surface. The peptide also degraded and reduced the biofilm formation in *C. parapsilosis* and *K. pneumonia* cells [59].

The antifungal activity of three peptides, called *Mo*-CBP3-PepI (CPIAQRCC) (#42), *Mo*-CBP3-PepII (NIQPPCRCC) (#43) and *Mo*-CBP3-PepIII (AIQRCC) (#44), designed based on the structure of *Mo*-CBP3, a chitin-binding protein purified from *Moringa oleifera* seeds, was evaluated against *C. albicans* and *C. parapsilosis* biofilms [60,61,76]. Eight SAMPs were tested regarding their antifungal potential against *C. neoformans,* and five SAMPs showed an inhibitory effect on *C. neoformans* growth at low concentrations. Peptides induced many morphological alterations, such as in the cell membrane, wall damage and loss of internal content in *C. neoformans* cells [56].

A designed and synthesized synthetic peptide consisting of 23 amino acids, named Octominin (1GWLIRGAIHAGKAIHGLIHRRRH23) (#45), from a defense protein 3 cDNA sequence of Octopus minor showed an inhibitory effect against *Candida albicans* by causing ultrastructural cell wall deformities [62].

Plants’ multifunctional protein osmotin plays an important role in plant immune systems, inducing abiotic stress tolerance and providing protection from fungal infections [46]. The cacao osmotin (#16)-like protein (TcOsm1)-derived peptides, named Osm-pepA (#46) and Osm-pepB, inhibited the growth of yeasts (*Saccharomyces cerevisiae* S288C and *Pichia pastoris* X-33) and spore germination of the phytopathogenic fungi *Fusarium f.* sp. *glycines* and *Colletotrichum gossypi.* Osm-pepA was more efficient than Osm-pepB [63].

Inspired by antimicrobial peptides, different side chain methionine, leucine, and tyrosine-based polymethacrylates (#51) (pH-responsive cationic biocompatible polymers) have been designed, and their fungicidal activities have been investigated against *A. niger*, demonstrating the effective inhibition of hyphal growth and distortion of conidiophores [77].

The antifungal activity of a series of peptides with a varying number of lysine and tryptophan residue repeats (KWn-NH_2_) was confirmed against *C. albicans* [78].

Summarizing the information presented, it can be concluded that various methods of the chemical modification of natural peptides, including phosphorylation, cyclization, halogenation, etc. [65], can be used to obtain semi-synthetic AFPs (Figure 4).

However, in addition to the same methods, computer molecular analysis and the design of the peptides being created play an important role in the design of synthetic AFPs, which makes it possible to proceed not from the natural basis that can only be modified, but from obtaining the synthesis of new compounds with the desired properties and taking into account the use of key amino acid motifs (Figure 4). At the same time, of course, all new AFPs must be evaluated not only for the effectiveness of the antifungal effect obtained from them but also for toxicity, including prion properties [79].

Thus, the use of various AFPs for their modifications described in this section is an effective approach to solving the problem of pathogenicity and resistance of fungi. The fact that these peptides, as natural molecules, make it possible to create a number of different biomimetics/peptidomimetics based upon them, which can be used as antifungals, allows us to conclude that there is a need for significant development of rational design methods today.

The use of organic synthesis methods makes it possible to construct SAFPs with the most effective amino acid sequences and desired characteristics, as well as changed mechanisms (targets) of action.

## 3. Effects of Different Biomimetics with Antifungal Activity of Peptide and Non-Peptide Nature on Fungi

In addition to AFPs, various molecules can act as biomimetics with antifungal action, which play a significant role in maintaining the viability and metabolic activity of fungal cells. Such molecules may include the potential involvement of non-canonical amino acids in the construction of structural proteins and enzymes, analogs of nitrogenous bases involved in the formation of nucleic acids (RNA and DNA), analogs of hormones involved in the formation of membrane structures, mimetics of QS molecules of fungi, etc.

The development of various materials and tools that acquire antifungal properties due to the inclusion of various biomimetics in their content is now coming to the forefront of scientific and practical research. At the same time, the action of biomimetics inside such developed materials can be divided into those that (Figure 1, Table 3, Table 4 and Table 5):-Prevent adhesion of spores and vegetative cells of fungi, as well as their subsequent invasive development;-Affect the membranes and cell walls of fungi, showing fungicidal activity, which leads to cell death;-Change the morphology and metabolic activity of fungal cells, inhibiting the synthesis of mycotoxins as a means of protecting fungi, as well as the most important enzymes of cells and QS molecules involved in the development of resistance of fungi to various influences;-Inhibit cell growth and spore formation, reducing the biosynthesis of adenosine triphosphate (ATP) and nucleic acids.

Further, a number of these main effects of different biomimetics on fungi will be considered separately. The antifungal substances of both peptide and non-peptide nature will be discussed in accordance with the targets of their action on fungal cells.

### 3.1. Biomimetics with a Destructive Effect on Fungal Cells

Polyene antibiotics are the most well-known, effective and widely used compounds with antifungal action. Nystatin (polyene macrolide synthesized by bacteria of the genus *Streptomyces*) is one of the classic antifungal antibiotics. However, there is a traditional interest in biomimetics, which are chemically modified nystatin molecules, the proposed mechanism of action of which is binding to ergosterol in the structure of lipid bilayers of fungal membranes. In this case, extra-membrane aggregates are formed that inhibit the transport of amino acids and glucose through the plasma membrane of cells [80]. Finally, polyene biomimetics contribute to the formation of macropores in the cell membrane and lead to their destruction.

The reason for the creation of new biomimetics is that nystatin itself has poor water solubility, instability to light, acidic air and heating, which limits its use. A significant amount of research is aimed at finding ways to avoid these disadvantages of natural nystatin. Chitosan, gelatin, alginate and other polymers are being studied for their modification by conjugation and improvement in the mentioned characteristics [81,82]. Selenium, bismuth and copper are often used to modify biomimetics based on polyene macrolides, which make it possible to enhance their antifungal effect [82,83,84].

An essential aspect of the ongoing research is the study of the possibility of increasing the specificity of the toxic effects of the obtained biomimetics on fungal cells. Thus, the obtained nystatin amides demonstrated increased water solubility and reduced toxicity in relation to human kidney cells (HEK293) while maintaining the inhibition of ergosterol biosynthesis by fungal cells [80].

In the case of another polyene macrocyclic antibiotic (amphotericin B) (**47**) with antifungal action, isolated from the extract of *Aloe vera* leaf, the effectiveness of such modification was shown, which leads to the formation of its nanoparticles with a slow release of the main substance, which causes the death of yeast cells and mycelial fungi [85,86] (Table 3 [17,77,86,87,88,89,90,91,92,93]). Often, the use of various biomimetics in the form of nanoparticles (often by encapsulating the main active substances) makes it possible to increase their antifungal activity. The in vitro investigation of the antifungal activity of itraconazole (#48) and difluorinated curcumin loaded within the chitosan nanoparticles showed a significant decrease in colony-forming units, as compared to free drugs [87].

Peptoids are considered as perspective peptidomimetics consisting of N-alkylated glycine oligomers (-RN-CH_2_-CO-) _n_ (#49). The compounds possess the capability of forming a wide variety of secondary structures, including α-helix, polyproline type I helix, and type I and type III β-turns. Peptoids have high resistance to proteolysis and demonstrate antifungal activity. It was revealed that peptoids cause disruption of the membrane and DNA molecules of *C. albicans* yeast cells [88].

**Table 3 biomimetics-08-00513-t003:** Biomimetics as antifungals with effect on fungal membranes and cell walls.

#	Biomimetics as Antifungals	Target of Action	* Antifungal Effect
47	Amphotericin B nano-aggregates [86]	Inhibition of ergosterol biosynthesis for membrane formation and provoking lysis of cells	MIC (mg/L): *Candida* spp. (0.125–0.5); *A. fumigatus* (1.0)
48	Itraconazole and difluorinated-curcumin containing chitosan nanoparticles in hydrogel [87]	Synergistic antifungal activity composed of increased permeation through fungal cell wall and membrane and lethal action of difluorinated curcumin	EC_50_ (mg/L): *Trichophyton mentagrophytes* (150)
49	*N*-alkylated glycine oligomers (peptoids) [88]	Suppressed formation of hyphae resulted in changes in cell and organelle morphology, most dramatically in the nucleus and nucleolus, and in the number, size and location of lipidic bodies	MIC (μM): *C. albicans* (3.0–13.0)
50	Synthetic peptides from predicted cysteine-rich peptides of tomato (mimicking γ-core regions of cysteine-rich peptides of *Solanum lycopersicum*) [89]	Charge of the derived peptides is positive, favoring interactions with the membranes of the pathogens. Inducing permeabilization and disruption of the fungal membranes	IC_50_ (μM): *Cryptococcus neoformans* (5.1–11.5); *Fusarium culmorum* (42.1–126.7); *F. oxysporum* (43.8–165.8); *F. solani* (47.5–138.8); *F. verticillioides* (99.8–152.0)
51	Methacryloyloxyethyl ester monomers with tyrosine, methionine and leucine [77]	Destruction of the cell membrane	MIC (mg/L): *A. niger* (0.16)
52	Biomimetic nanopillar Si-containing surfaces [17]	Rupture of coat and inner membrane of spores leading to cell death	A 4-fold reduction of amounts of attached spores and approximately 9-fold reduction of viable conidia of *Aspergillus brasisiensis* on biomimetic surface
53	Chitosan/polyethylene oxide (CPO) [90]	Antifungal effect similar to voriconazole (production of intracellular ROS and increasing permeability of plasma membrane)	*C. albicans* cells: diameter of inhibition area by CPO—25–27 mm (agar disc diffusion method); Control: inhibition by voriconazole—27 mm
54	Phe-Ala dipeptide polymer/polyoxometalate composite [91]	Deformation of conidial heads and the appearance of indistinguishable sterigmates; smooth cell walls of hyphae become completely depressed and destroyed; spores become wrinkled	MIC (mg/L): *A. niger* (230)
55	Metal-organic framework (Ce-MOF) with enzyme-like activity of catalase, superoxide dismutase and peroxidase [92]	Production of ROS and inhibition of fungal growth	40 mg/L Ce-MOF provides 93.3–99.3% growth inhibition of *Aspergillus flavus*, *A. niger*, *A. terreus*, *C. albicans*, *Rhodotorula glutinis*
56	Iodine-doped carbon dots (IDCDs) with peroxidase-like activity [93]	Production of ROS	The 90% decrease in number of *C. albicans* CFU is observed in the presence of 2.72 g/L IDCDs and 0.5 mM H_2_O_2_ under visible light irradiation over 2 h

* MIC, minimum inhibitory concentration; EC_50_, half maximal effective concentration.

Synthesis of the biomimicks based on γ-core regions of cysteine-rich peptides of *Solanum lycopersicum* (#50) allowed the obtaining of antifungal compounds that permeabilize fungal membranes of yeast-like fungus *Cryptococcus neoformans*, fungi of the genera *Fusarium*, *Bipolari*s and *Botryti*s [89].

The fungicidal activity was established in a number of biomimetic pH-responsive amphiphilic polymers based on the amino acids found in antimicrobial peptides (tyrosine, methionine, leucine) (#51) synthesized via the reversible addition–fragmentation chain transfer polymerization technique. Such polymers violated the structural integrity of hyphae. Treatment with cationic polymers induced the distribution of sterol components within the plasma membrane lipid raft, followed by inhibition of the growth of hyphae.The interactions of fungal cells with polymers subsequently have prompted the altered cellular signaling resulting in the failure of polarized cell growth. [77]. As a result, there was a complete loss of cellular integrity due to the emerging membrane depolarization and subsequent cell lysis. Among all the amino acids, polymers with methionine showed higher activity than with leucine and tyrosine.

Interestingly, from the point of view of antifungal effects, was the development of various materials—contact with which leads to a decrease in the number of living fungal cells. It was shown that by enhancing hydrophobicity or hydrophilicity, as well as changing the topography of the surface of materials into which, for example, nanostructured Si is introduced (#52), it is possible to simulate the surface of dragonfly wings [17]. Such biomimicking materials possess antifungal activity due to the prevention of attaching *A. brasiliensis* fungal spores and conidia, and the contact of fungal cells with such materials results in the rupture of the coat and inner membrane of spores, leading to cell death.

New biomimetic nanofibers that are based on chitosan with polyethylene oxide (#53) acting against *C. albicans* have been developed. The antifungal effect was similar to known voriconazole [90]. The integration of peptides into synthetic macromolecular systems is attractive for obtaining materials with antifungal properties. Thus, the molecular assembly of a cationic polymer based on peptides with anionic polyoxometallates leads to the formation of a well-defined structure.

A composite based on Phe-Ala dipeptide polymer and Keggin phosphotungstate (#54) was developed. Two amino acids, phenylalanine and alanine, were chosen to produce the dipeptide because of their amphiphilic properties and their ability to form higher-order self-assembling structures in water [91]. The resulting composite showed fungicidal activity against *A. niger* cells by increasing the local charge in the polymer. A significant deviation in cellular integrity and a complete alteration in cell morphology was observed. The branches of mycelia became flattened and stalked together, while no presence of conidiophores was noticed.

The cerium-based organometallic framework (Ce-MOF) (#55) turned out to be very interesting because of its high (93.3–99.3%) effectiveness in inhibiting the growth of vegetative cells of *A. flavus*, *A. niger*, *A. terreus*, *C. albicans* and *Rhodotorula glutinis* due to the generation of a high concentration of ROS [92]. Ce-MOF also caused extensive (up to 77.4%) deformation of the conidiophores of fungi with spores. The antifungal activity of Ce-MOF was due to its ability to act as a biomimetic of several oxidative enzymes at once (catalase, superoxide dismutase and peroxidase). The observed morphological changes consisted of the violation of the integrity of the cell walls of fungi and the release of cell contents as a result of the effect of ROS generated by Ce-MOF on the cells.

Iodine-doped carbon dots (IDCDs) (#56) were synthesized as peroxidase mimics for the antifungals against *Candida albicans*. The antifungal activity of the IDCDs was 90% in the presence of exogenous H_2_O_2_ (0.5 mM) and under visible light irradiation for 2 h. The IDCDs promote the formation of ROS, including hydroxyl radicals (OH), which can successfully inhibit the growth of *C. albicans* [93]. This material imitated the action of the peroxidase used for yeast disinfection. In relation to the enzyme, the peroxidase-like action of this biomimetic material was more effective due to the presence of iodine in its content.

Despite the great interest in biomimetics that causes the death of fungal cells, there remains a general problem with their practical application, which is associated with the need for their toxicity to other eukaryotic cells [94]. The search for the most highly specific active substances with a minimum level of toxicity for non-fungal cells remains relevant.

### 3.2. Biomimetics with Inhibition Effect on Fungal Proteins/Enzymes

To inhibit the enzymatic activity of fungi, synthetic substrates acting as mimetics or specific inhibitors can be used to reduce the metabolic activity of fungi or to carry out their catalytic or physiological functions, reducing the biosynthesis of a number of metabolites and preventing their growth (Table 4) [16,95,96,97,98,99,100,101,102]. Biomimetic substrates should, on the one hand, inhibit the widest possible range of fungal enzymes and, on the other hand, not be inhibitors for enzymes of plant and animal origins.

**Table 4 biomimetics-08-00513-t004:** Biomimetics as antifungals with effect on fungal proteins/enzymes.

#	Biomimetics as Antifungals	Target of Action	* Antifungal Effect
57	Sulfonyl hydrazide derivatives containing the 1,2,3,4-tetrahydroquinoline [95]	Inhibition of laccase activity	EC_50_ (mg/L): *Sclerotinia sclerotiorum* (3.32), *Valsa mali* (2.78)
58	Iridoid alkaloids biomimic of camptothecin [96]	Strong inhibitory effects against mycelial growth and spore germination. Disturbing the replication and transcription of DNA by binding to topoisomerase I, inhibiting ergosterol biosynthesis	LC_50_ (mg/L): *F. graminearum* (34.5); *Rhizoctonia solani* (18.0); *Botrytis cinerea* (26.0)
59	Hymexazol glycosides [16]	Inhibition of chitinase, produced by fungi	EC_50_ (mg/L): *Alternaria alternata* (1.58)
60	Synthetic oxime-derivatives of resorcylate aminopyrazole [97]	Selective inhibition of chaperone Hsp90 functions	EC_50_ (μM): *Candida neoformans* (0.040); *C. albicans (*0.011)
61	pCF2Ser peptide as substrate mimetic [98]	Inhibition of Cdc14 phosphatases; formation of defective conidiation and ascospores, reducing cell virulence	Inhibitory constant (Ki) against fungal phosphatase Cdc14 homologs: 3–19 µM
62	*N*-1-(β-d-ribofuranosyl) benzimidazole derivatives [99]	Inhibition of fungal cytochrome P_450_ 3A-dependent C14-α-demethylase, which is responsible for the conversion of lanosterol to ergosterol and ergosterol biosynthesis	MIC (mg/L): *A. flavus* (0.8); *A. niger* (1.6); *F. oxysporum* (3.1); *C. albicans* (0.8)
63	3-aryl-isoquinoline derivatives [100]	Inhibition of succinate dehydrogenase activity	EC_50_ (mg/L): *Physalospora piricola* (3.7)
64	Dextran-coated Gd-based nanoparticles (NPs) as phosphatase-like nanozyme [101]	Selective hydrolysis of the terminal high-energy phosphate bonds in ATP	464 mg/L NPs increase ethanol yield by 17%. The characteristics of NPs: Km, Vmax and Ea were 29.4 μM, 7.17 × 10^−7^ M/s and 29.34 kJ/mol, respectively
65	Nanopillars of poly(methyl methacrylate)-like cicada wing surface topography [102]	Superhydrophobic surfaces with reduced adsorption capability of proteins needed for adhesion of fungal spores	100% removal of 10^5^ spores of *Fusarium oxysporum* on the surface with antifungal activity

* MIC, minimum inhibitory concentration; EC_50_, half maximal effective concentration.

New fungicides (#57) based on 4-chlorocinnamaldehyde theosemicarbazide (PMDD-5Y) and 1,2,3,4-tetrahydroquinoline that inhibited fungal laccases were obtained. Antifungal activity of the new compounds was confirmed against *Sclerotinia sclerotiorum* and *Valsa mali* cells [95].

The biomimetic synthesis of iridoid alkaloids (#58) resulted in the obtaining of compounds with antifungal activity similar to camptothecin, which is a topoisomerase inhibitor, a natural quinoline alkaloid. These iridoid alkaloids (#58) disrupted DNA replication and transcription by binding to topoisomerase I, and then interrupted mycelial cell division and inhibited ergosterol biosynthesis [96].

Hymexazol (HYM) (#59) can inhibit the spore germination of fungi by combining aluminum and iron ions or can interfere with the RNA or DNA biosynthesis of pathogenic fungi. Modification of HYM (#59) by amino sugars significantly stimulates plant growth but causes morphological changes in the mycelium *Alternaria alternata*: plasma membranes are separated from cell walls, cytoplasm becomes vacuolized, and mitochondria, lysosomes and other organelles decrease in size in cells, while the mechanical strength of the cell walls is significantly reduced. Glycosylated HYM (#59) suppresses the activity of chitin synthase G involved in chitin biosynthesis, which leads to these changes in cells and limits their growth [16].

Imidazole is a structural element of many important biological compounds: the essential amino acid histidine, the purine bases within the structures of nucleic acids (DNA and RNA), and ATP. Biomimetics obtained as derivatives of imidazole-containing compounds have an inhibitory effect on many chemical reactions in living cells. Many of these compounds are widely distributed in cells and predominantly in a protein-bound state [103].

Such interactions lead to the inhibition of bound proteins and enzymes. Namely, inhibition of the ergosterol synthesis by direct binding to sterol 14α-demethylase (CYP51) is the main function of the imidazole derivatives as antifungals. The membrane-binding enzyme CYP51 is localized in the outer membrane of the endoplasmic reticulum and catalyzes the removal of the methyl group at the fourteenth carbon atom [104]. Recently, there has been a decrease in the effectiveness of the inhibitory effect of antifungals based on azoles in relation to CYP51 due to mutational changes in the enzyme [105]; however, the synthesis of new derivatives continues.

Among the biomimetics obtained as imidazole derivatives, many compounds are known today that contain several halogen atoms (F or Cl) to enhance their antifungal effect [106]. For example, fluconazole and albaconazole contain two F-atoms, whereas itraconazole contains two Cl atoms.

Synthetic aminopyrazole-substituted resorcylatamides (#60) have been studied as inhibitors of the functioning of the chaperone protein Hsp90 [97]. Hsp90’s function supports protein quality control mechanisms, productive folding and the stability of conformationally labile proteins, many of which are involved in key signaling cascades. The chaperoning by Hsp90 of its so-called client proteins is ATP-dependent and coordinated by a suite of cochaperones and accessory factors that impart client selectivity and help regulate progression through the chaperoning cycle. The Hsp90 isoform is based on the semi-synthetic oxime-derivatization of the resorcylate macrocycle natural products radicicol (1) and monocillin I (a substance synthesized by the fungi *Monocillium nordinii* and other microorganisms). The modification made it possible to obtain selective inhibitors of fungal chaperones that have minimal negative effects on human chaperones.

To inhibit fungal phosphatase Cdc14, which is a key regulator of mitosis in yeast and fungal pathogens, a peptide substrate mimetic (#61) containing a non-hydrolyzable α,α-difluoromethylene phosphonoserine (pCF2Ser) residue instead of pSer was developed and investigated [98]. The inhibition constant (Ki) for the new pCF2Ser peptide (#61) was determined for a number of Cdc14 homologues present in the *Saccharomyces cerevisiae*, *Cercospora zeina*, *Magnaporthe oryzae*, *Rhizoctonia solani* and *Puccinia striiformis* cells, which depended on the cell type, and was in the range of 3–19 µM. It was found that *Fusarium graminearum* and *Aspergillus flavus* fungus cells devoid of active Cdc14 did not form conidia and ascospores and could not infect plants [98].

Lanosterol 14α-demethylase is a key enzyme in the biosynthesis of ergosterol in fungi. This enzyme is a target for the action of new antifungals. New *N*-1-(β-d-ribofuranosyl) benzimidazole derivatives (#62) were obtained, and their fungicidal activity against several pathogenic fungal strains was evaluated (*Aspergillus flavus*, *A. niger*, *F. oxysporum* and *C. albicans*) [99]. Maximum inhibitory efficacy was found against *A. flavus* and *C. albicans*, which turned out to be better than that known for the fungicide fluconazole against the same cells. It is interesting to note that the hydrophilicity of the resulting compound (#62) was the determining factor for enhanced antifungal activity, while hydrophobic compounds showed increased antibacterial activity.

Succinate dehydrogenase provides electrons for aerobic respiratory circuits in the mitochondria of eukaryotic fungal cells. New 3-aryl-isoquinoline derivatives (#63), based on sanguinarine (heleritrine and berberine) inhibiting this enzyme, have been synthesized [100]. The fungicidal activity of the compounds (#63) obtained at a concentration of 50 mg/L was higher in comparison with the antimicrobial action of sanguinarine, which inhibits mycelial growth of *Alternaria solani*, *A. alternata* and *Physalospora piricola*. Molecular docking showed that the obtained compounds (#63) completely overlap the active center of succinate dehydrogenase, interacting with negatively charged amino acid residues of chain A of this enzyme (Thr58, Ala61, Glu398, Leu415, Leu418, etc.), and thus inhibit the enzyme, significantly affecting the metabolism of fungal cells.

Nanoparticles (#64) of phosphatase-like nanozyme, based on gadolinium covered by dextran, were synthesized as mimetic of enzymatic catalysts to intensify the dephosphorylation of ATP, an important biomolecule reflecting the energy status of living cells [107]. The use of dextran made it possible to increase the biocompatibility and endocytosis of nanoparticles (#64) in yeast cells. The resulting samples selectively catalyzed the hydrolysis of terminal high-energy phosphate bonds in the ATP molecule, led to a decrease in the concentration of this important substance in the yeast cells of *S. cerevisiae* and increased the yield of ethanol during glucose fermentation [101].

In fact, there was a stimulation of the transition of cells from active growth under aerobic conditions to glycolysis under anaerobic conditions with a rapid decrease in growth rate. A decrease in the concentration of intracellular ATP and the accumulation of an increased concentration of ethanol in the medium with cells resulted in the inhibition of cell metabolism.

It is known that *Aspergillus fumigatus* and *F. oxysporum* fungi form spores that are transported with air masses and adhere to various biological surfaces through protein–protein and protein–peptide interactions involving membrane-bound fungal proteins. Studies of fungicidal properties of biomimetic-nanostructured surfaces made of poly(methyl methacrylate) (#65) with characteristics close to a “cicada wing” in relation to these fungi have shown that such materials prevent the adhesion of spores and their germination with the subsequent development of fungal infections [102]. The same materials (#65) proved to be effective against the adhesion of ascospores of *S. cerevisiae* yeast cells.

Thus, such materials counteract the adhesion of fungal proteins and prevent them from attaching to bio-surfaces.

### 3.3. Biomimetics of Metabolites with Effects on Growth and Metabolic Activity of Fungi

Various fungal metabolites become important targets of antifungal exposure [108]. The use of the created biomimetics is aimed at their competition with the substituted molecules, which affects the metabolic activity and growth of fungal cells [109]. The well-known metabolomics of fungi enables the development of approaches to decrease fungal resistance and adaptation to various stress conditions (ionizing radiation, pH change, hypoxic stress, solar ultraviolet radiation, presence of toxic compounds, nutrient stress, oxidative stress, cold–heat stress) (Table 5) [18,110,111,112,113,114,115,116,117,118].

**Table 5 biomimetics-08-00513-t005:** Biomimetics as antifungals with effect on the growth and metabolic activity of fungi.

#	Biomimetics as Antifungals	Target of Action	* Antifungal Effect
66	4-fluorophenylalanine (FPA) [110]	Incorporation in proteins and inhibition of cell growth	Twofold decrease in growth rate of *Sacharomyces serevisiae* by 500 mg/L FPA
67	Formyl phloroglucinol meroterpenoids [111]	Reducing hyphae elongation and filamentation due to blocking of potential outflow of fungal substrates	MIC_50_ (mg/L): *C. albicans* (8.7), *C. glabrata* (13.5)
68	2-(2-hydroxypropyl) phenol [112]	Inhibition of respiration, causing a decrease in ATP concentration and metabolic activity	EC_50_ (μg/mL): *Rhizoctonia cerealis* (1.0); *Pythium aphanidermatum* (20.3); *V. mali* (14.9); *Botrytis cinerea* (23.5)
69	Fraxinellone [113]	Changes in lipopolysaccharide-induced DNA-binding activity and reduced translation	EC_50_ (mg/L): *Alternaria longipes* (64.2); *Curvularia lunata* (123.3)
70	l-pyroglutamic acid 4-chiral hydroxyl sulfonyl ester derivatives [18]	Inhibition of biosynthesis of trichothecenes	61.6% inhibition of *Fusarium graminearum* growth by 100 mg/L
71	2-deoxyglucose [114,115]	Violation of glycolysis and ATP biosynthesis	>2 times increase in the doubling time of *S. cerevisiae* cells
72	1-amino-1-(4-imidazole)methylphosphonic acid [116]	Inhibition of various enzymes, especially proteases	MIC (mg/L): *Rhodotorula mucilaginosa* (1024), *A. niger* (5000)
73	{[(2-hydroxy-4-nitrophenyl)amino](thiophen-3-yl)methyl}phosphonic acid (5N3TPA) and {[(2-hydroxy-4-methylphenyl)amino](thiophen-3-yl)methyl}phosphonic acid (5M3TPA) [117]	Inhibition of fungal enzymes	78.42% and 50% inhibition of *Fusarium oxysporum* and *Alternaria alternata* growth by 100 mg/L of 5N3TPA and 5M3TPA, respectively
74	Oxidized α-pheromone [118]	Oxidation abolishes chemoattractant activity and quorum-sensing activity of α-pheromone	Reduced inhibition of *Fusarium oxysporum* spore germination by ~1.5 times

* MIC, minimum inhibitory concentration; EC_50_, half maximal effective concentration.

In recent decades, considerable attention has been paid to the development of antimicrobial and antitumor agents in the form of natural and non-canonical amino acid derivatives [119,120]. It has been established that such unnatural amino acids can be incorporated by yeast cells into their proteins, which leads to a noticeable inhibition of cell growth (Table 5) [110]. It was shown that up to 35–40% of phenylalanine can be replaced by its fluorine-containing analog (#66) in the cell biomass of yeast *Saccharomyces cerevisiae*. The introduction of amino acids with halogen atoms to the media with fungi results in the biosynthesis of peptides and proteins with their electron-withdrawing features and non-natural steric effects because the fluorination influences the folding properties, changing the secondary structure propensity of peptides and protein segments containing aliphatic halogenated amino acids. In addition, halogen atoms have an impact on the proteolytic stability of the peptides [121].

One of the other approaches in the molecular design of antifungal active substances is associated with the introduction of nitrogen-containing heterocyclic fragments into the structure of compounds. The tetrazole cycle is one of the most promising in this context. NH-disubstituted tetrazoles are sufficiently strong acids, comparable in acidity to carboxylic acids and rather weak Brønsted bases. High metabolic stability and the ability to overcome biological membranes ensure the successful use of the tetrazole cycle as a bioisoster of some functional groups in biologically active compounds. Tetrazolyl analogs of L-glutamic acid, L-tyrosine and L-ornithine as potential amino acid biomimetics have been synthesized and characterized.

Under environmental conditions, halogenated amino acids are synthesized by some fungi and require enzymes to introduce the halogen to protein or peptide structures within a post-translational modification. Non-proteinogenic amino acids may have diverse physiological functions and regulate the functioning of metabolic cycles. The properties of halogenated AMPs, both native and synthetic, and their possible contributions to the growing problem of antibiotic resistance of microorganisms are currently being investigated [121].

Formyl phloroglucinol meroterpenoids are characteristic metabolites of the plant genus *Eucalyptus* and exhibit fungicidal activity against various *Candida* species, including strains resistant to fluconazole. Terpenoids were found to be potential efflux pump substrates of fungi. Their fungicidal activity is enhanced by conjugating them with triphenyl phosphonium cations aimed at mitochondria. In this study, terpenoids were modified with phloroglucinol groups (#67), and it was shown that the compounds obtained are most active against *Candida albicans* and *C. glabrata*—13.5) and are comparable in efficacy with fluconazole. In addition, the compounds obtained at concentrations of 25 and 100 mg/L suppressed the growth of yeast biofilms by 66% and 90%, respectively [111].

2-Allylphenol (#68) is a synthetic fungicide that mimics ginkgo from *Ginkgo biloba*, effectively used against various fungal pathogens (*V. mali*, *Botrytis cinerea*, *Drechslera turcica*, *Rhizoctonia ceramicis* and *Sphaerotheca aphanis*). The antifungal mechanism of action includes a decrease in ATP biosynthesis and inhibition of cell respiration. As a result of such an impact, protoplasm outflow occurs from hyphae, which, after a few hours, begin to collapse [112].

Natural limonoid derivatives with antifungal activity can be isolated from plant sources and used against the pathogenic fungus *Cladosporium cucumerinum* to reduce the rate of fungal growth [122]. Fraxinellone (#69) is the most effective antifungal compound from this group [123]; however, this compound and many other lemonoid derivatives can be obtained in a semy-synthetic or synthetic process [113,124]. The synthesis of potential antifungals represents the biomimetics of natural limonoids based on the natural enzyme-catalyzed conversion of squalene to lanosterol with the participation of oxidosqualene cyclase and squalene epoxidase (monooxidase). Lanosterol is a tetracyclic triterpenoid and is the compound from which all fungal steroids are derived.

Secondary metabolites of plants in the form of terpenoids, alkaloids and various organic acids can exhibit fungicidal properties. Thus, l-pyroglutamic acid, isolated from the perennial herbaceous plant *Disporopsis aspersa*, demonstrates an antifungal effect against *Phytophthora infestans* and *Psilocybe cubensis*. l-pyroglutamic acid is an analog and potential precursor of glutamate, which is necessary for the biosynthesis of glutathione. l-pyroglutamic acid reduces the accumulation of trichothecenes in fungi of the genus *Fusarium*. Studies of the expression of genes that induce the biosynthesis of trichothecenes have shown that the production of mycotoxins is inhibited by l-pyroglutamic acid at the transcription level. A number of derivatives of l-pyroglutamic acid from L-hydroxyproline were synthesized. It turned out that the reaction of l-pyroglutamic acid with 4-chlorophenol to form an ester (#70) enhances its antifungal activity. Among the 31 synthesized compounds at a concentration of 100 mg/L, the maximum inhibition level (61.6%) of the fungal growth of *F. graminearum*, comparable to the commercial fungicide chlorothalonil, were derivatives obtained by the introduction of naphthalene or alkane by hydroxyl group. The electron-withdrawing groups on the aromatic ring facilitated the inhibitory effect [18].

In studies of fungal metabolism, structural analogs of substrates are used, which makes it possible to “close” one of the possible metabolic pathways for the study of alternative pathways. Thus, as a result of the studies of glycolysis and related processes in yeast cells, it was found that deoxyglucose (2-DG) (#71), imitating glucose, as well as mannose, changes not only the rate of glycolysis but also many other processes (the glycosylation of proteins, ATP production and the regulation of signaling pathways), causing the energy depletion of cells and generation of stress in them [114]. As a result, the rate of cell doubling is reduced by more than 2 times. Despite the fact that in the course of such studies it was found that yeast cells consuming 2-DG (#71) are able to mutate and develop resistance to this substrate [115], the use of 2-DG (#71) as an inhibitor of yeast metabolism can be further investigated under certain conditions, in particular, in combinations with other antifungals.

Phosphonates are compounds with diverse biological activity, including antimicrobial ones [125,126]. The aminophosphonates, being structural analogs of natural amino acids, inhibit the activity of various enzymes, especially fungal proteases [116,126]. Due to this property, aminophosphonic acids containing heterocyclic functional groups are used as pharmaceutical preparations or pesticides. However, the presence of a heterocycle reduces the toxicity of these compounds compared to aromatic hydrocarbon analogs [116]. Recently, it was revealed that a number of enzymes of different microorganisms are capable of the degradation and transformation of similar compounds [125,127]; since then, further possible use of these compounds can also be considered in the presence of other antifungals in combinations with them to improve the results.

Currently, various aminophosphonic acids containing a thiophene ring (#72) are actively synthesized, which exhibit antifungal activity against fungi and yeast cells [116,117]. The presence of active functional groups in the aromatic ring, the P=O and P-OH groups of α-aminophosphonates and the thiophene ring enhances the antifungal activity of these compounds. Molecular docking of newly synthesized aminophosphonic acids (#72) confirmed the possibility of a notable inhibition of proteases.

Similar to other ascomycete fungi, *F. oxysporum* secretes an α-pheromone (#73), a small peptide that functions as a chemoattractant and as a QS molecule. Three of the ten amino acid residues of the α-pheromone are tryptophan residues composed of a side chain with a high affinity for lipid bilayers. The use of lipid mimetics based on phospholipids (analogs of the cell membrane bilayer) allows the binding of this QS molecule and the formation of an intramolecular disulfide bond between two cysteine residues during the interaction. It should be noted that the oxidized version of the α-pheromone (#73) had no biological activity as a chemoattractant or QS molecule [118]. This effect leads to a weakening of the QS, which plays an important role in the functioning of fungal and yeast populations [128] and ensures a more effective action of the antimicrobial agents used simultaneously.

Thus, the processes that make up the metabolism of filamentous fungi and yeast and the study of fungal metabolomics can serve as a source of new ideas for creating effective antifungals. Substances regulating the metabolic activity of microbial cells, including end-products of their metabolism, can significantly affect the survival and energetic status of fungal cells. The largest number of antifungals being developed in this area relate to inhibitors of one or another stage of cell metabolism.

## 4. Combination of Antifungal Peptides with Each Other and/or with Antifungal Drugs

Despite expectations of the absence of resistance formation in fungi to the effects of new antifungals, including AFPs [129,130], azoles [131] and 2-DG [114], researchers note such cases. One of the effective solutions to this problem today is the combined use of various antifungals. Such possible combinations can include various traditional antifungal agents and AFPs (Table 6) [19,20,39,59,76,132,133,134,135,136,137,138,139,140,141,142,143,144,145]. Such combinations make it possible to join different mechanisms (targets) of the effects of antifungals and significantly decrease their toxicity by improving the effectiveness of the action and reducing the doses applied in comparison with individual substances.

The synergistic effect of combined antifungals makes it possible to effectively combat yeast biofilms of the genus *Candida*, both at the stage of their formation and already for the degradation of mature biofilms. A combination of AFPs *Mo*-CBP_3_-PepI (#42) and *Mo*-CBP_3_-PepIII (#44) with nystatin (#75) and itraconazole (#48) against *Candida* species biofilms resulted in a 2- to 4-fold improvement of antibiofilm activity of antifungals [59,76]. The combination of *Mo*-CBP_3_-PepIII (44) and *Rc*Alb-PepIII (#76) synthetic peptides with itraconazole (#48) enhanced the activity of the latter by 10-fold against *C. neoformans* [132]. Likewise, synthetic histidine-containing amphipathic peptides (#78,80) enhanced the activity of amphotericin B (#47) against *Cryptococcus neoformans* by 4- to 16-fold [20,133]. When combined with lactoferrin-derived synthetic peptide lactofungin (#81), the minimum inhibitory concentration of amphotericin B (#47) for *Candida* spp. and *C. neoformans* decreased by 4 times [134].

The antifungal activity of two *N. fischeri* peptides, NFAP (#11) and NFAP2 (#82), and their γ-core peptide derivatives (γ^NFAP^-opt, γ^NFAP2^-opt) (#38), was tested in vitro against *Botrytis*, *Cladosporium and Fusarium* spp. A synergistic mechanism of action was observed when NFAP (11) or NFAP2 (#82) was applied in combination with γ^NFAP^-opt (#38). The investigated proteins and peptides did not show any toxicity to tomato plant leaves, except for γ^NFAP2^-opt [135].

*Fusarium* infections have been associated with high mortality rates due to the lack of effective treatment strategies. The in vitro activity of AFPs MSI-78 (#36), hLf(1-11) (#24) and cecropin B (#15) combined with amphotericin B (#47) or voriconazole (#83) were tested against ten *Fusarium solani* strains. All AFPs demonstrated a synergistic mechanism of action when combined with conventional antifungals [39].

The combination of brilacidin (a synthetic, nonpeptidic, small molecule mimetic of defensin) (#84) with  caspofungin (#21) has a synergism that is able to affect *A. fumigatus* viability through multiple mechanisms of action, encompassing functional changes/depolarization of the microorganism cell membrane, interference to calcineurin signaling, and misexpression of the cell wall integrity pathway, and prevents β-1,3-glucan biosynthesis [136].

The use of mimetic peptides DP-23 (short lipopeptide) (#86) and SPO (N-substituted polyglycine) (#87) in combination with fluconazole (#79) provided a synergistic effect on *A. niger* and *A. flavus* cells [18].

Peptides P255 and P256 (#88), obtained from hexapeptide PDF 26, were combined with amphotericin B (#47) and investigated against *C. albicans*. Both peptides showed a synergistic effect with a polyene antibiotic, while P256 showed a stronger antifungal effect than P255. Peptides violated the integrity of the cell wall, increased membrane permeability, disrupted cell morphology and caused intracellular changes: they affected the expression of genes for replication and repair of fungal DNA, the biosynthesis of cell wall components and ergosterol. They also increased the production of ROS in cells and bound to the genomic DNA of fungi [137].

γ-AA peptides are a class of peptidomimetics with a definite folded structure and resistance to proteolytic hydrolysis. It has been shown that lipo-γ-AA peptide MW5 (#89) significantly increases the efficacy of fluconazole (#79) action against azole-resistant *C. albicans* CARG5 cells. The peptide destroys the cell membrane and provokes the production of ROS [138].

Synthetic peptidomimetics of antimicrobial α-peptides exhibit fungicidal activity as a result of the hydrophobic and electrostatic interactions with cell membranes, which lead to permeabilization and subsequent cell death. These peptidomimetics were more stable than their natural counterparts and were not degraded by proteases. β-peptides (#90) were obtained as structurally constructed on the basis of natural α-helical AMPs. These mimetics (#90) possessed a fungicidal effect against *C. albicans* cells and inhibited the formation of yeast biofilms. The hydrophobicity of the β-peptides directly correlates with their antifungal properties and a narrow range of concentrations at which β-peptides effectively kill *C. albicans* cells without lysis of erythrocytes. The combination of these peptides (#90) with isoamyl alcohol (#91) reduced their MIC for inhibiting biofilms by 4 times [139].

Conjugates of fluoroquinolone antibiotics (ciprofloxacin or levofloxacin) (#93) with the TP10-NH_2_ peptide (#92) penetrating into yeast cells showed an antifungal effect against different strains of genus *Candida*. At the same time, the synergistic effect of these conjugates was not revealed in the experiments on *C. glabrata* cells [140].

Antifungal effect against *Candida* spp. cells was investigated for conjugates obtained on the basis of a modified fragment of lactoferrin HLopt2 (#94) and ciprofloxacin, levofloxacin (#93) and fluconazole (#79). Three different nutrient media were used for these purposes. There was no activity against four different *Candida* species for the substances studied on the RPMI (mimics physiological conditions) medium. The use of BP (1% peptone) and BHI1/100 (0.034% brain-heart infusion) media resulted in the inhibition of cell growth, especially *C. tropicalis*, whereas *C. glabrata* cells were the most resistant among all tested strains [141].

The antifungal activity of cationic antimicrobial peptides ToAP2 (#95) (from a cDNA library of the scorpion *Tityus obscurus* venom gland) and NDBP-5.7 (#96) (from a cDNA library of the scorpion *Opisthacanthus cayaporum* venom gland) was demonstrated against *C. albicans* cells. Both peptides affected membrane permeability and caused such changes in the morphology of fungal cells as cell wall deformations and disruption of ultrastructural cell organization. Both peptides showed synergism with amphotericin B (#47) and demonstrated synergistic and additive effects in combination with fluconazole (#79), TopAP2 (#95) and NBDP-5.7 (#96), respectively [142].

The unique ultra-short peptide KW23 (#97) had a positive effect on standard and resistant *Candida* species, showing powerful synergistic antimicrobial activity in combination with fluconazole (#79). Interestingly, was the fact that the effect of this combination was additive with respect to the resistant strain. In addition, the peptide showed a low enough toxicity to human erythrocytes [143].

Persistent cells that can tolerate lethal concentrations of antimicrobials and grow again after their removal are found in different microbial populations. It is assumed that the appearance of persistent yeast cells involves different regulations of genes correlating with the pathways of ergosterol biosynthesis (ERG1 and ERG25) and β-1,6-glucan (SKN1 and KRE1). Special studies have shown that membranes of cells in biofilms contain a lower concentration of ergosterol, mainly in the deepest layers of the biofilm, in comparison with planktonic yeast cells. It is likely that cells from mature biofilms require less ergosterol to maintain membrane fluidity, and this is also confirmed by the limited effectiveness of biomimetics aimed at inhibiting ergosterol biosynthesis. In this regard, the possibility of destroying resistant persistent-derived biofilms of *C. albicans* with gH625-M peptide (#98), which is an analog of the viral membranotropic peptide gH625, was investigated [144]. The combination of gH625-M (#98) with various antifungals (fluconazole (#79), 5-flucytosine (#99) and amphotericin B (#47)) demonstrated a synergistic effect when acting on persistent cells in biofilms using relatively low doses of the antifungals (Table 6).

Among the antifungals that can be used in effective combination with AFPs, metal-containing compounds and enzymes are discussed [146,147]. The main mechanism of the antifungal action of metals is the triggering of the generation and accumulation of ROS. Enzymes are of special interest due to their wide range of substrate specificity of action and various mechanisms of antifungal actions.

The combination of metalloenzymes, such as hexahistidine-containing organophosphorus hydrolase (His_6_-OPH), capable of hydrolyzing signal molecules of yeast QS, with AFPs (polymyxins (#2,3), bacitracin, Lfcin (#5)) resulted in a notable improvement in the antimicrobial efficiency of action of the AFPs (up to 8.5 times) against different yeast species, such as *Saccharomyces cerevisiae*, *Candida* sp., *Trichosporon beigeii*, etc. It is interesting that His_6_-OPH had an increased catalytic efficiency of action in the hydrolysis of its substrates (QS molecules) when it was introduced in combination with AFPs [148,149]. Thus, the combination of the AFPs with enzymes possessing antifungal activity is a perspective trend in the development of efficient antifungals.

To determine the effect of combining two antifungals and interpreting the results obtained, a fractional inhibitory concentration index (FICI) range (from 0.5 to 4) is usually applied. The synergistic effect corresponds to FICI values of < 0.5 and is achieved through a combination of different mechanisms of action on antifungals. From the studies analyzed in this section, it follows that only in the case of a combination of fengycin (#100) with surfactin (#101) that an antagonistic effect was revealed (FICI = 4), which is an indicator of the incompatibility of the mechanisms of action of these compounds with each other [145].

In the case of a combination of brilacidin (84) with conventional fungicides, for AFPs of *Neosartorya fischeri* (#11,38,82), among themselves and histidine-containing amphipathic peptides (#78,80) with fluconazole (#79), an additive effect (FICI > 1) was marked, which means that these antifungals do not enhance the effectiveness of each other’s actions [133,135,136]. In the case of all the other analyzed combinations, a synergistic effect was revealed. The most effective variants were obtained in the case of combinations of synthetic peptides Mo-CBP3-PepI (#42) and Mo-CBP3-PepII (#43) with nystatin (#75) (FICI = 0.13) [59,76]. It is interesting to note that when combining two synthetic peptides, MSI-78 (#36) and hLf(1-11) (#24), with a natural peptide, Cecropin B (#15), as well as with voriconazole (#83) or with amphotericin B (#47), the strongest synergistic effect was achieved with a natural peptide; however, not with considered successful antifungals like azole and polyene antibiotic [39].

When developing biomimetics and some combinations with them, special attention is paid to minimizing their toxicity (or its complete absence) in relation to human/animal/plant cells, as well as to the type of antifungal effect (mechanism of action) on cells in order to avoid the development of resistance [80,150]. At the initial stage of such investigations, toxicity is assessed using modern computer modeling methods [151]. Erythrocytes are used as experimental models both in vitro and in vivo [152].

Taking into account the fact that a minimal number of cases of fungal resistance have been detected with respect to SAFPs, as well as a minimum level of toxicity shown in these peptides for other cells, they seem promising candidates in the development of modern antifungals.

## 5. Conclusions

Analyses of the data in Table 1 and Table 2 showed that the molecular weight of SAFPs is often comparable to the molecular weight of natural AFPs—it may be more/less than it; however, the structure rather than the weight of the substances affects their antifungal properties. Most natural phases turn out to be effective at slightly higher concentrations compared to SAFPs.

Discussing the concentration ranges recommended for the use of biomimetics, it should be noted that it is quite difficult to compare this indicator for different groups of drugs since, in some studies, different indicators of their effectiveness are used (MIC, LC_50_, EC_50_, and up (Table 1, Table 2, Table 3, Table 4, Table 5 and Table 6)) and are presented in different units of measurement. Nevertheless, when trying to assess the concentrations of different biomimetics necessary for their antifungal effect, we consider that it is possible to conditionally arrange the groups of preparations in the following row with increasing compared parameters: Synthetic/(semi)synthetic AMPs with antifungal activity < biomimetics as antifungals with effects on fungal membranes and cell walls < natural AMPs with antifungal activity < biomimetics as antifungals with effects on fungal proteins/enzymes < biomimetics as antifungals with effects on the growth and metabolic activity of fungi. To estimate the effectiveness of the newly developed antifungals, widespread test cultures are mainly used, depending on the intended application; for example, yeast of the genus *Candida*, mycelial fungi of the genus *Aspergillus* or *Fusarium* (Table 1, Table 2, Table 3, Table 4, Table 5 and Table 6).

From this perspective, new SAFPs are expected to appear with a wider range of antifungal effects, as well as peptides that can have a targeted and selective effect on the cells of potentially dangerous fungal cultures for humans and animals, as well as on fungal phytopathogens, such as, for example, as representatives of the genera *Acremonium*, *Absidia*, *Paecilomyces* or *Scopulariopsis*. A great potential of specifically acting antifungal agents can be disclosed through proteomic studies of fungal pathogens and the search for key proteins and enzymes—the inhibition of which can provide a maximum and rapid effect without the formation of fungal resistance [153]. The creation of combinations of peptides and new antifungal agents may soon yield significant results for impact not only on resistant strains but also on heterogeneous biofilms [143,144]. Moreover, specific combinations can reduce the toxicity of individual components in relation to human cells [141,143].

Already developed combinations of SAFPs with other antifungal agents, as it turns out, can effectively affect resistant strains of fungi, even those which are in the state of QS (biofilms). Such combined antifungals seem to be a worthy solution to one of the important health and food safety problems associated with the resistance of fungi to the antimicrobials used.

## Figures and Tables

**Figure 1 biomimetics-08-00513-f001:**
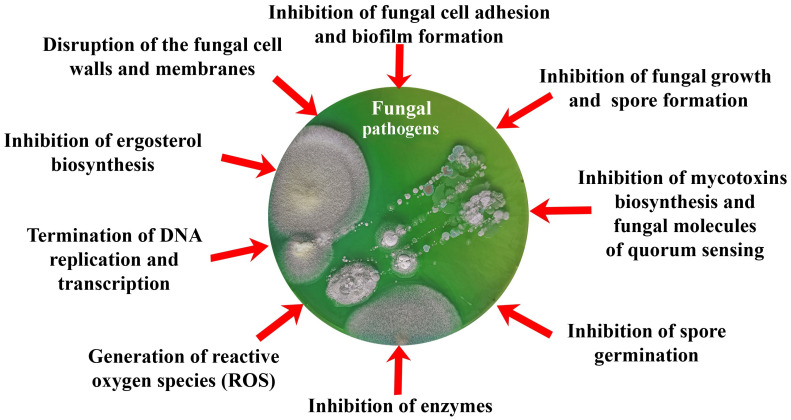
Current trends in the development of biomimetic antifungals analyzed in this review.

**Figure 2 biomimetics-08-00513-f002:**
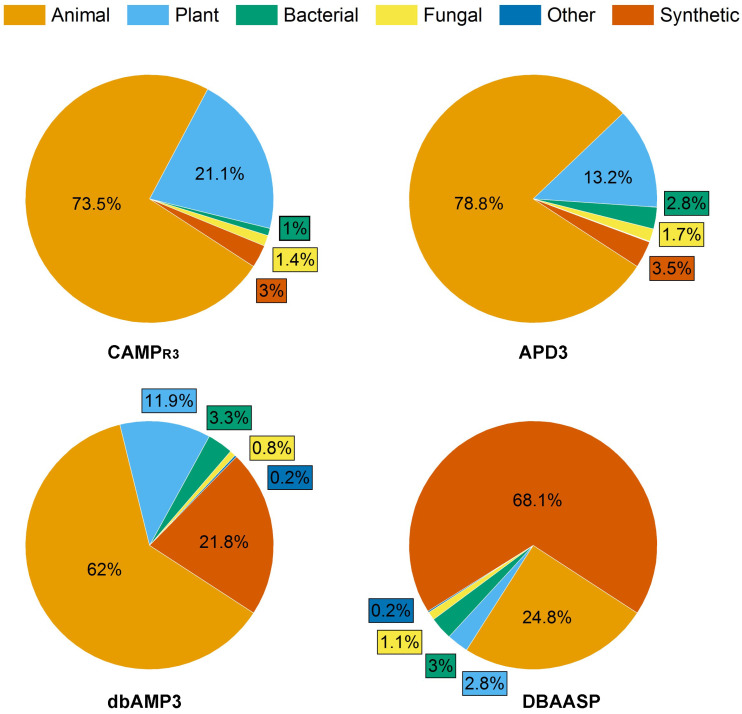
Composition of AMP databases according to the source of peptides with antifungal activity. Total number of AFPs in each database was taken as 100%: 1311, 1144, 5968 and 5540 for APD3, CAMP_R3_, DBAASP and dbAMPs, respectively.

**Figure 3 biomimetics-08-00513-f003:**
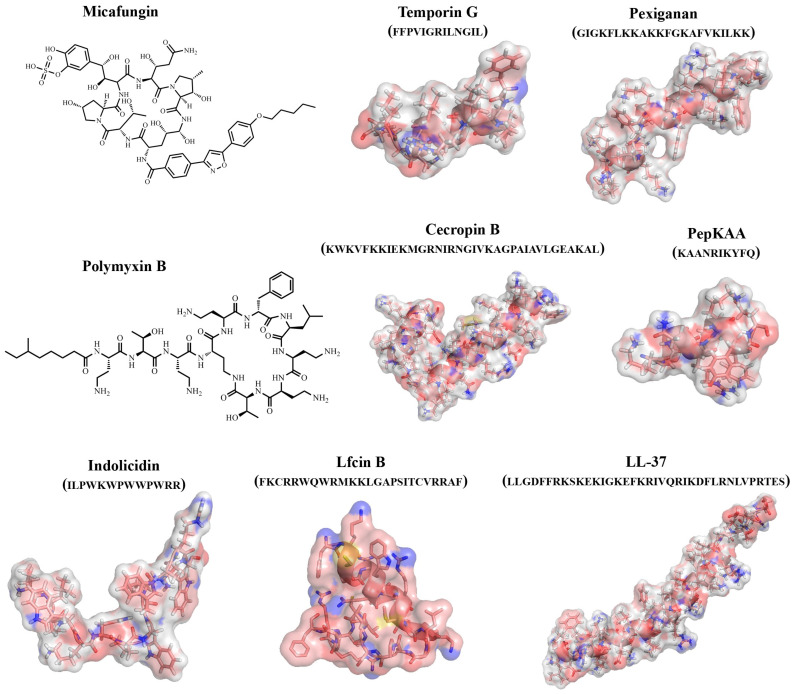
Examples of structures of natural and synthetic AFPs.

**Figure 4 biomimetics-08-00513-f004:**
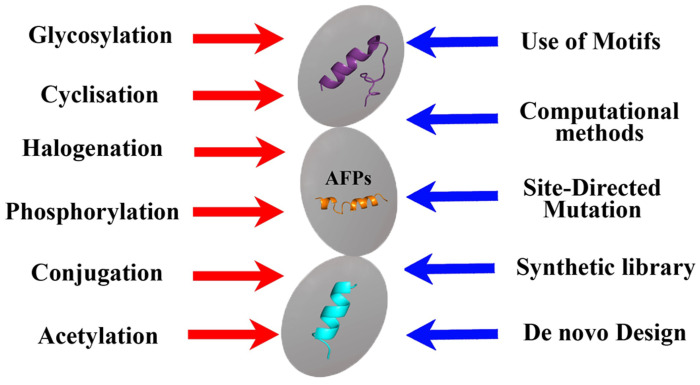
Approaches used in the development of biomimetic AFPs.

**Table 1 biomimetics-08-00513-t001:** Natural AMPs with antifungal activity.

#	Antifungal Peptide	Origin	Molecular Weight [Reference]	Antifungal Effect
**Mechanism of action: inhibition of chitin biosynthesis**				
1	Nikkomycin Z	fungi	495.4 Da [27]	0.5–64 mg/L (yeasts, fungi) ^a^
**Mechanism of action: destabilization of plasma membrane, pore formation, cell wall damage**				
2	Polymyxin B	bacteria	1301.6 Da [28]	16–256 mg/L (multi-drug-resistant fungal strains) ^a^
3	Colistin	bacteria	1155.4 Da [28]	
4	Lactoferampin B	bovine	2389.8 Da [29]	0.7–39 µM (*C. albicans*) ^c^
5	Lactoferricin B	bovine	3125.8 Da [29]	0.31–400 mg/L (yeasts) ^a^ 4–32 µM (fungi) ^a^
6	Lactoferricin H	human	5513 Da [29]	10 mg/L (*C. albicans*) ^a^
7	Halictine Hal-2	sweat bee	1452.85 Da [30]	1.6–25 µM (*Candida* spp.) ^a^
8	Halocidin	ascidian	3445.1 Da [31]	Effect was not estimated
9	Magainin-2	frog	2466.9 Da [32,33]	6.25 µM (*Saccharomyces cerevisiae*, *Trichosporon beigelii*, *Candida albicans*) ^a^ 60-100 mg/L (*Penicillium digitatum*; *Alternaria solani*; *Phytophthora infestans*) ^a^
10	Defensin DefMT3	ticks	1613.1 Da [34]	4 µM (*Fusarium culmorum*; *F. graminearum*) ^b^
11	*Neosartorya fischeri* antifungal protein (NFAP)	ascomycete	6600 Da [35]	12.5–100 mg/L (fungi) ^a^
12	Indolicidin	bovine	1906.3 Da [36]	12.5–50 mg/L (*C. albicans*) ^a^
13	Leg2	chickpea legumin hydrolysates	2157.6 Da [37]	125–250 µM (*S. cerevisiae*, *Zygosaccharomyces bailii*) ^a^
14	LL-37	human	4493.3 Da [38]	4–64 µM (*Candida* spp.) ^a^
**Mechanism of action: cell/spore lysis, cell wall perturbations**				
15	Cecropin B	silkworm	3835.7 Da [39]	0.9 mg/L (*C. albicans*) ^a^ 160–320 mg/L (*F. solani*) ^a^
16	Osmotin	plant	24285.3 Da [40]	4–25 mg/L (fungi) ^a^
17	Stomoxyn	stable fly	4474.2 Da [41]	0.8–50 µM (yeasts); 0.4–7 µM (fungi) ^a^ 50–100 µM (*A. fumigatus*) ^a^
18	Temporin B	frog	1391.8 Da [42]	1.4–4 µM (*Candida* spp.) ^a^
19	Temporin G		1457.8 Da [43]	8–128 µM (yeasts/fungi) ^a^

^a^ MIC, minimal inhibiting concentration; ^b^ IC_50_, half maximal inhibitory concentration; ^c^ LC_50_, 50% lethal concentration.

**Table 2 biomimetics-08-00513-t002:** Synthetic/(semi)synthetic antimicrobial peptides with antifungal activity.

#	Antifungal Peptide	Molecular Weight; [Reference]	Antifungal Effect
**Mechanism of action: inhibition of 1,3-β-d-glucan synthase**			
20	Anidulafungin	1140.2 Da [44,45,46]	0.06–0.25 mg/L (*Candida* spp.) ^a^ 0.015–32 mg/L (fungi) ^a^
21	Caspofungin	1093.3 Da [44,45,46]	0.25–4 mg/L (*Candida* spp.) ^a^
22	Micafungin	1270.3 Da [44,45,46]	0.015–4 mg/L (*Candida* spp.) ^a^
23	CGA-N12-0801	993.2 [47]	3.5–31.4 mg/L (*C. tropicalis*) ^a^
**Mechanism of action: destabilization of plasma membrane, pore formation, cell wall damage**			
24	Lf(1-11) H	1317.5 Da [39]	>12.5 mg/L (*Candida* spp.); 80–160 mg/L (*F. solani*), 4.3 µM (*A. fumigatus*) ^a^
25	Lfchimera (bLfcin/Lfampin)	4422 Da [29]	6.25 mg/L (*C. parapsilosis*) ^a^
26	PPD1	<1000 Da [48,49]	~4.9 mg/L (*A. flavus)* ^a^
27	66-10		~4 mg/L (*A. flavus)* ^a^
28	77-3	994.2 Da [48,49]	3.5–5 mg/L (*A. flavus*, *A. parasiticus)* ^a^
29	D4E1	2079.4 Da [48]	7.75 µM (*A. flavus*); 0.60 µM (*V. dahliae*) ^b^ 13.02 µM (*C. destructivum*) ^b^
30	KK14	144.2 Da [50]	6.25–100 mg/L (fungi) ^a^
31	PAF26	991.2 Da [51]	4–6 µM (*P. digitatum*) ^a^
32	C12O3TR	n.d. [52]	3.12–25 mg/L (fungi, yeasts) ^a^
33	Trp-His[1-(3,5-di-tert-butylbenzyl)]-NHBn	n.d. [53]	3.81 mg/L (*C. neoformans*) ^a^
34	Halictine Hal-2 derivatives	1471 Da [54]	0.5–1 µM (*Candida* spp., *S. cerevisiae*) ^a^
35	di-K19Hc	4115.1 Da [55]	<4 mg/L (*C. albicans*); <16 mg/L (*Aspergillus* sp.) ^a^
36	Pexiganan/MSI-78	2478.2 Da [39]	10–80 mg/L (*F. solani*) ^a^
37	γ-core DefMT3	1611.8 Da [34]	1–2 µM (*F. culmorum*; *F. graminearum*) ^b^
38	γ^NFAP^, γ^NFAP^-opt	1700 Da [35]	12.5–200 mg/L (fungi) ^a^
39	PepGAT	1044.18 Da [56,57,58]	40–80 mg/L (*Candida* spp., *P. digitatum*) ^a^
40	PepKAA	1238.44 Da [56,57,58]	
41	*Rc*Alb-PepII	637.77 Da [56,59]	17–250 µM (*Candida* spp.) ^a^ 0.04 mg/L (*Cryptococcus neoformans*) ^a^
**Mechanism of action: production of reactive oxygen species, cell wall degradation**			
42	*Mo*-CBP_3_-PepI	893.12 Da [60]	2.2 µM (*C. albicans*) ^c^
43	*Mo*-CBP_3_-PepII	1031.30 [60]	17.5 µM (*C. albicans*) ^c^
44	*Mo*-CBP_3_-PepIII	692.85 [61]	
45	Octominin	2652.2 Da [62]	50 mg/L (*C. albicans*) ^a^
**Mechanism of action: Cell/spore lysis, cell wall perturbations**			
46	Osm-pepA	3050.5 Da [63]	40 µM (*S. cerevisiae*) ^a^ 20 µM (*Pichia pastoris*) ^a^

^a^ MIC, minimal inhibiting concentration; ^b^ IC_50_, half maximal inhibitory concentration; ^c^ IC_90_, concentration that inhibited yeast growth by 90%.

**Table 6 biomimetics-08-00513-t006:** Combination of AFPs with different antifungal agents.

Components of Combined Antifungals		Target [Reference]	Antifungal Effect
**(#) AFP as the first component**	**(#) Second component**		
(#42) *Mo*-CBP_3_-PepI	(#75) Nystatin	*C. albicans* [59]	0.13 ^a^
(#43) *Mo*-CBP_3_-PepII			
(#44) *Mo*-CBP_3_-PepIII		*C. parapsilosis* [76]	82% ^b^
	(#48) Itraconazole		96% ^b^
(#76) *Rc*Alb-PepIII		*C. neoformans* [132]	84.1% ^b^
(#77) l-His(2-adamantyl)-l-Trp-l-His(2-phenyl)-OMe	(#47) Amphotericin B	*C. neoformans* [20]	0.28 ^a^
(#78) l-Trp-l-His(1-biphenyl)-NHBzl		*C. neoformans* [133]	0.28 ^a^
	(#79) Fluconazole		1.04 ^a^
(#80) l-His[1-(4-n-butylphenyl)]-l-Trp-l-His[1-(4-n-butylphenyl)]-NHBzl	(#47) Amphotericin B		0.31 ^a^
	(#79) Fluconazole		0.75 ^a^
(#81) Lactofungin	(#47) Amphotericin B	*C. albicans*, *C. glabrata*, *C. neoformans*, *C. deuterogattii* [134]	0.16–0.28 ^a^
(#11) Neosartorya fischeri AFPs (NFAP)	(#82) NFAP2	*Botrytis cinerea*, *Cladosporium herbarum* [135]	1.25 ^a^
(#11) NFAP	(#38) γ^NFAP^-opt		0.28–1.50 ^a^
(#82) NFAP2	(#38) γ^NFAP^-opt		0.31–1.5 ^a^
(#36) MSI-78	(#83) Voriconazole	*Fusarium solani* [39]	0.34 ^a^
(#24) hLf(1-11)			0.21 ^a^
(#15) Cecropin B			0.17 ^a^
(#36) MSI-78	(#47) Amphotericin B		0.37 ^a^
(#24) hLf(1-11)			0.31 ^a^
(#15) Cecropin B			0.28 ^a^
(#84) Brilacidin (non-peptide mimetic of host defense peptides)	(#21) Caspofungin	*Aspergillus fumigatus* [136]	0.39 ^a^
	(#83) Voriconazole		1.0 ^a^
	(#85) Geldanamycin		0.64 ^a^
(#86) DP-23 peptoid	(#79) Fluconazole	*A. flavus*, *A. niger* [19]	0.16–0.38 ^a^
(#87) SPO peptoid			
(#88) P256 and P256	(#47) Amphotericin B	*C. albicans* [137]	0.28 ^a^
(#89) γ-AA peptide MW5	(#79) Fluconazole	*C. albicans* [138]	≤0.5 ^a^
(#90) 14-helical β-peptide	(#91) Isoamyl alcohol	*C. albicans* [139]	4 mg/L ^d^
(#92) TP10-NH_2_ (analog of transporan 10)	(#93) Ciprofloxacin or Levofloxacin	*Candida* spp. [140]	6.3–100 μM ^c^
(#94) HLopt2 (mimic of human lactoferrin)	(#79) Fluconazole	*Candida* spp. [141]	2–125 mg/L ^c^
(#95) ToAP2	(#96) NDBP-5.7	*C. albicans* [142]	0.75 ^a^
(#95) ToAP2	(#47) Amphotericin B		0.18 ^a^
(#96) NDBP-5.7			0.18 ^a^
(#95) ToAP2	(#79) Fluconazole		0.5 ^a^
(#96) NDBP-5.7			0.56 ^a^
(#97) KW-23		*C. albicans* [143]	0.37–0.60 ^a^
(#98) gH625M		*C. albicans* [144]	0.30 ^a^
	(#99) Flucytosine		0.20 ^a^
(#98) gH625M	(#47) Amphotericin B		0.5–0.8 ^a^
(#100) Fengycin	(#101) Surfactin	*Rhizopus solonifer* [145]	5 ^a^

^a^ Fractional inhibitory concentration index (FICI). FICI values suggest antagonistic (>4.0), indifferent (>1 to <4), additive (>0.5–1) and synergistic (≤0.5) effects. ^b^ Percent of biofilm and growth inhibition; ^c^ MIC/MIC_50_, minimum inhibitory concentration (complete inhibition/at least 50% inhibition); **^d^** minimal biofilm prevention concentrations (MBPC).

## Data Availability

Not applicable.

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
