# Peer review of "Various Biomimetics, Including Peptides as Antifungals"

_biomimetics, 2023, doi:10.3390/biomimetics8070513_

Round 1
Reviewer 1 Report
Comments and Suggestions for Authors
The manuscript by Elena Efremenko etal reviewed semi-synthetic and synthetic antifungal peptides. This review systematizes the current scientific information, developing various antifungals with assessment of the effectiveness of the created biomimetics for antimicrobial anti-fungal effects.
This review is very interesting.
I suggest to publication after a major revision.
Major point:
1. The review mainly involves the research progress of several peptides in antimicrobial aspects, and the title should reflect peptides.
2. The article mentions a variety of synthetic and semi-synthetic peptides, but there is no structures, the authors should add the structures of typical peptides.
Comments on the Quality of English Language
Minor editing of English language required Minor editing of English language required.
Author Response
Dear Reviewer,
We are grateful to you for the suggestions allowing us the improving of our manuscript.
Please, see our comments to your remarks and the revised text of the paper:
Responses to the Comments of Reviewer:
Major point:
- The review mainly involves the research progress of several peptides in antimicrobial aspects, and the title should reflect peptides.
Thank you for your suggestion. We modified the Title of the manuscript.
- The article mentions a variety of synthetic and semi-synthetic peptides, but there are no structures, the authors should add the structures of typical peptides.
Thank you for your suggestion. We added Figure 3 with the examples of different structures of peptides with antifungal properties to Section 2 of the review text.
With kind regards,
The authors of the manuscript

Reviewer 2 Report
Comments and Suggestions for Authors
The manuscript submitted by Efremenko et al. is a review describing a number of semisynthetic and synthetic antifungal substances of various types. The topic is undoubtedly hot. There is still a growing number of papers on antifungals. According to Web of Science, the number of reviews with the word "antifungal" in the title grew from ca. 30-50 in years 2000-2016 to more than 140 in 2022. Thus, it seems necessary to cite the most relevant ones (e.g., 'Natural and synthetic peptides with antifungal activity', Future Medicinal Chemistry Vol. 8, No. 12, https://doi.org/10.4155/fmc-2016-0035 or 'Cyclic Peptides with Antifungal Properties Derived from Bacteria, Fungi, Plants, and Synthetic Sources', Pharmaceuticals 2023, 16(6), 892, https://doi.org/10.3390/ph16060892 - these are examples only, I am sure the authors know perfectly which reviews should be cited) and provide an explanation, in which aspects the manuscript under the review is unique and different from the previous ones.
The authors should also define what type of antifungals are included in the manuscript. Aimed at health? Food? Agriculture? Wood? Buildings? Or various ones?
In the first part of the review, antifungal peptides (AFPs) are discussed. In Table 1, 18 natural AFPs are listed, while in Table 2, 20 semisynthetic and synthetic AFPs are shown. Since in Fig. 1 there are several thousands of AFPs, the question arises which criteria were used to choose so few AFPs in the Tables?
The organization of material presented is somehow chaotic:
1. Not all antifungals in Tables 1-6 are discussed in following texts, and vice versa, not all discussed antifungals are listed in the tables.
2. The order of substances in tables and descriptions is different. It should be of the same order. Tables should have an additional column for entry numbers (preferentially continuous along all tables), and these entries should be referred to in the description.
3. The first part of section 2 'Antifungal peptides (AFPs)' is rather general; then there is a more detailed part with examples. I suggest adding a subtitle after line 157.
4. Section 3 is entitled 'Biomimetics with antifungal activity of non-peptide nature'. Despite that there are several substances of peptide nature included there:
· Table 3: Methacryloyloxyethyl ester monomers with tyrosine, methionine and leucine
· Table 3: Synthetic peptides from predicted cysteine-rich peptides of tomato (mimicking γ-core regions of cysteine-rich peptides of Solanum lycopersicum)
· Table 3: N-alkylated glycine oligomers (peptoids)
· Three paragraphs on lines 344-364
· The paragraph on lines 379-386
· Table 4: pCF2Ser peptide as substrate mimetic
· Table 5: 4-fluorophenylalanine (FPA)
· Table 5: Phe-Ala dipeptide polymer/polyoxometalate composite.
They should be moved to Section 2.
These issues require corrections and reorganization of the manuscript.
Minor errors noted:
· l. 39-42: The sentence 'Most of them are synthetic compounds, since then there are concerns about their high toxicity due to limited water solubility and the need to use their relatively high concentrations, as well as about the decrease in the effectiveness of their action due to the emerging resistance in fungi' is illogical. After 'Most of them are synthetic compounds, since then there are…' one expects advantages, not disadvantages.
· Throughout the text: in 'NH2' the digit '2' should be in subscript.
· l. 176, 244, 248; Table 5 (entry '2- deoxyglucose'): remove spaces before and after the hyphen
· Table 4 & l. 475: change '-d-ribo…' into '-D-ribo…'; 'D' should be a small cap
· Throughout the text: the stereochemical descriptors 'L' & 'D' should be small caps
· l. 484: change 'aril' to 'aryl'
· Table 5: remove spaces in {[(2-hydroxy-4-nitrophenyl) amino] (thiophen-3-yl) methyl} phosphonic acid (should be: {[(2-hydroxy-4-nitrophenyl)amino](thiophen-3-yl)methyl}phosphonic acid) and {[(2-hydroxy-4-methylphenyl) amino] (thiophen-3-yl) methyl} phosphonic acid (should be: {[(2-hydroxy-4-methylphenyl)amino](thiophen-3-yl)methyl}phosphonic acid)
· l. 612: change 'compared to aromatic analogues' to 'compared to aromatic hydrocarbon analogues'. Heterocycles may be (and often are) aromatic as well.
· l. 1018: add a dot and a space before 'doi'
Comments on the Quality of English Language· Throughout the text: generally, there should be no comma before the word 'that'. Remove it in l. 164, 170, 174, 480, 532, 756, and maybe, in other instances omitted by the reviewer.
· l. 164: change '…relation to cells Candida spp.' to '…relation to cells of Candida spp.'
· l. 734 change 'removal, being found' to 'removal are found'
Author Response
Dear Reviewer,
We are grateful to you for the suggestions allowing us the improving of our manuscript.
Please, see our comments to your remarks and the revised text of the paper:
Responses to the Comments of Reviewer:
The manuscript submitted by Efremenko et al. is a review describing a number of semisynthetic and synthetic antifungal substances of various types. The topic is undoubtedly hot. There is still a growing number of papers on antifungals. According to Web of Science, the number of reviews with the word "antifungal" in the title grew from ca. 30-50 in years 2000-2016 to more than 140 in 2022. Thus, it seems necessary to cite the most relevant ones (e.g., 'Natural and synthetic peptides with antifungal activity', Future Medicinal Chemistry Vol. 8, No. 12, https://doi.org/10.4155/fmc-2016-0035 or 'Cyclic Peptides with Antifungal Properties Derived from Bacteria, Fungi, Plants, and Synthetic Sources', Pharmaceuticals 2023, 16(6), 892, https://doi.org/10.3390/ph16060892 - these are examples only, I am sure the authors know perfectly which reviews should be cited) and provide an explanation, in which aspects the manuscript under the review is unique and different from the previous ones.
We updated the list of references and cited a few most relevant articles on antifungal peptide [Ciociola, T.; Giovati, L.; Conti, S.; Magliani, W.; Santinoli, C.; Polonelli, L. Natural and synthetic peptides with antifungal activity. Fut. Med. Chem. 2016, 8, 1413-1433. doi: 10.3389/fcimb.2020.00105; Rautenbach, M.; Troskie, A.M.; Vosloo, J. A. Antifungal peptides: To be or not to be membrane active. Biochimie 2016, 130, 132-145. doi: 10.1016/j.biochi.2016.05.013; Li, T.; Li, L.; Du, F.; Sun, L.; Shi, J.; Long, M.; Chen, Z. Activity and Mechanism of Action of Antifungal Peptides from Microorganisms: A Review. Molecules 2021, 26, 3438. doi: 10.3390/molecules26113438; Helmy, N.M.; Parang, K. Cyclic Peptides with Antifungal Properties Derived from Bacteria, Fungi, Plants, and Synthetic Sources. Pharmaceuticals 2023, 16, 892. https://doi.org/10.3390/ph16060892] (references #15,64,65,67). We would also like to draw your attention to the equally interesting articles which were already cited by us in the original review text [Fernández de Ullivarri, M.; Arbulu, S.; Garcia-Gutierrez, E.; Cotter, P.D. Antifungal peptides as therapeutic agents. Front. Cell. Infect. Microbiol. 2020, 10, 105. doi:10.3389/fcimb.2020.00105; Struyfs, C.; Cammue. B.P.A.; Thevissen, K. Membrane-interacting antifungal peptides. Front Cell Dev Biol. 2021, 9, 649875. doi: 10.3389/fcell.2021.649875; Huan, Y.; Kong, Q.; Mou, H.; Yi, H. Antimicrobial Peptides: Classification, design, application and research progress in multiple fields. Front. Microbiol. 2020, 11, 582779. doi: 10.3389/fmicb.2020.582779] (references #14,63,66).
The difference of our review from others is that it contains information not only about antifungal peptides/proteins, but also about other substances that exhibit antifungal activity. In addition, we discuss combining - we lead the reader to the idea of combination and, when choosing objects to present in the review, we were guided by the demonstration of the wide potential of antifungal compounds. Not limited to peptides, but also focused the reader on the possibility of combining these compounds, including peptides, which are very suitable for this because of their properties, enzymes that act on various fungal targets, and with metals that generate reactive oxygen species.
The authors should also define what type of antifungals are included in the manuscript. Aimed at health? Food? Agriculture? Wood? Buildings? Or various ones?
In our article we considered various antifungals with a wide range of possible application. Since, as we have mentioned in the Introduction, the fungal pathogenesis is a common problem it is necessary to consider antifungals aimed at different targets and area of application.
In the first part of the review, antifungal peptides (AFPs) are discussed. In Table 1, 18 natural AFPs are listed, while in Table 2, 20 semisynthetic and synthetic AFPs are shown. Since in Fig. 1 there are several thousands of AFPs, the question arises which criteria were used to choose so few AFPs in the Tables?
As it was mentioned by the Reviewer, today a huge number of peptides are known and covering everything in one article is nearly impossible task. When selecting substances for discussion in our review, we were guided by the number of studies conducted with each of the peptides, the presence of a known structure, as well as the presence of synthetic analogues in the case of the natural peptides considered. The main idea was to show how appropriate it is to obtain certain synthetic analogues of natural peptides and how does these change the peptide properties (antimicrobial efficiency, toxicity, stability, mechanism of action, etc.). At the same time, the most recent and relevant synthetic peptides obtained by prediction and studied against various pathogens were also considered.
The organization of material presented is somehow chaotic:
- Not all antifungals in Tables 1-6 are discussed in following texts, and vice versa, not all discussed antifungals are listed in the tables.
It was corrected.
- The order of substances in tables and descriptions is different. It should be of the same order. Tables should have an additional column for entry numbers (preferentially continuous along all tables), and these entries should be referred to in the description.
We corrected the structure of tables and the text in descriptions so that they are of the same order. We also added explanation in description in order to make it clearer and easier to follow. Since adding the entry numbers to tables and text can lead to confusion in the structure and confuse the reader, we believe that listing the names of substances in the order they appear in the table is more rational.
- The first part of section 2 'Antifungal peptides (AFPs)' is rather general; then there is a more detailed part with examples. I suggest adding a subtitle after line 157.
Thank you for your suggestion. We added a subtitle “2.1 Examples of synthetic analogues of the natural AFPs” to section 2.
- Section 3 is entitled 'Biomimetics with antifungal activity of non-peptide nature'. Despite that there are several substances of peptide nature included there:
- Table 3: Methacryloyloxyethyl ester monomers with tyrosine, methionine and leucine
- Table 3: Synthetic peptides from predicted cysteine-rich peptides of tomato (mimicking γ-core regions of cysteine-rich peptides of Solanum lycopersicum)
- Table 3: N-alkylated glycine oligomers (peptoids)
- Three paragraphs on lines 344-364
- The paragraph on lines 379-386
- Table 4: pCF2Ser peptide as substrate mimetic
- Table 5: 4-fluorophenylalanine (FPA)
- Table 5: Phe-Ala dipeptide polymer/polyoxometalate composite.
They should be moved to Section 2.
The substances of peptide nature you mentioned are inserted into other tables in accordance with their effect on fungi (target for action). Therefore, transferring them to Table 1 is not consistent with the logic and structure of our review article, which will lead to inconvenience for the reader. We have a slightly original view on the content of the review - to collect not so much substances by their chemical nature, but by the point of impact on the object. At the beginning we write about peptides, since this is a traditional object in research against fungi, but then we say that peptides and other substances can be used to influence different targets. In order to make it clearer and do not confuse the readers we have edited the title of the Section 3 to “Effects of different biomimetics with antifungal activity of peptide and non-peptide nature on fungi”.
These issues require corrections and reorganization of the manuscript.
Minor errors noted:
- l. 39-42: The sentence 'Most of them are synthetic compounds, since then there are concerns about their high toxicity due to limited water solubility and the need to use their relatively high concentrations, as well as about the decrease in the effectiveness of their action due to the emerging resistance in fungi' is illogical. After 'Most of them are synthetic compounds, since then there are…' one expects advantages, not disadvantages.
- Throughout the text: in 'NH2' the digit '2' should be in subscript.
- l. 176, 244, 248; Table 5 (entry '2- deoxyglucose'): remove spaces before and after the hyphen
- Table 4 & l. 475: change '-d-ribo…' into '-D-ribo…'; 'D' should be a small cap
- Throughout the text: the stereochemical descriptors 'L' & 'D' should be small caps
- l. 484: change 'aril' to 'aryl'
- Table 5: remove spaces in {[(2-hydroxy-4-nitrophenyl) amino] (thiophen-3-yl) methyl} phosphonic acid (should be: {[(2-hydroxy-4-nitrophenyl)amino](thiophen-3-yl)methyl}phosphonic acid) and {[(2-hydroxy-4-methylphenyl) amino] (thiophen-3-yl) methyl} phosphonic acid (should be: {[(2-hydroxy-4-methylphenyl)amino](thiophen-3-yl)methyl}phosphonic acid)
- l. 612: change 'compared to aromatic analogues' to 'compared to aromatic hydrocarbon analogues'. Heterocycles may be (and often are) aromatic as well.
- l. 1018: add a dot and a space before 'doi'
Thank you for your recommendations. It was corrected.
Comments on the Quality of English Language
- Throughout the text: generally, there should be no comma before the word 'that'. Remove it in l. 164, 170, 174, 480, 532, 756, and maybe, in other instances omitted by the reviewer.
- l. 164: change '…relation to cells Candida spp.' to '…relation to cells of Candida spp.'
- l. 734 change 'removal, being found' to 'removal are found'
Thank you for your recommendations. It was corrected.
With kind regards,
The authors of the manuscript.

Round 2
Reviewer 1 Report
Comments and Suggestions for Authors
The authors addressed all issues we concerned.
Comments on the Quality of English LanguageMinor editing of English language required
Author Response
Dear Reviewer,
Thank you for your comments. We have checked our text with the aim of minor editing of the English language and made the necessary corrections.
With best wishes,
Authors of the manuscript.
Reviewer 2 Report
Comments and Suggestions for Authors
The authors partially improved the manuscript. The answers to my queries seem to be reasonable, but these explanations should be included in the manuscript. It is the readers, not the reviewer. who should know the area of the review. It is the readers, not the reviewer, who should know why some AFPs are discussed in section 2, while some other in subsequent sections.
Thus, the explanations:
"The difference of our review from others is that it contains information not only about antifungal peptides/proteins, but also about other substances that exhibit antifungal activity. In addition, we discuss combining - we lead the reader to the idea of combination and, when choosing objects to present in the review, we were guided by the demonstration of the wide potential of antifungal compounds. Not limited to peptides, but also focused the reader on the possibility of combining these compounds, including peptides, which are very suitable for this because of their properties, enzymes that act on various fungal targets, and with metals that generate reactive oxygen species."
" In our article we considered various antifungals with a wide range of possible application. Since, as we have mentioned in the Introduction, the fungal pathogenesis is a common problem it is necessary to consider antifungals aimed at different targets and area of application."
"As it was mentioned by the Reviewer, today a huge number of peptides are known and covering everything in one article is nearly impossible task. When selecting substances for discussion in our review, we were guided by the number of studies conducted with each of the peptides, the presence of a known structure, as well as the presence of synthetic analogues in the case of the natural peptides considered. The main idea was to show how appropriate it is to obtain certain synthetic analogues of natural peptides and how does these change the peptide properties (antimicrobial efficiency, toxicity, stability, mechanism of action, etc.). At the same time, the most recent and relevant synthetic peptides obtained by prediction and studied against various pathogens were also considered."
"The substances of peptide nature you mentioned are inserted into other tables in accordance with their effect on fungi (target for action). Therefore, transferring them to Table 1 is not consistent with the logic and structure of our review article, which will lead to inconvenience for the reader. We have a slightly original view on the content of the review - to collect not so much substances by their chemical nature, but by the point of impact on the object. At the beginning we write about peptides, since this is a traditional object in research against fungi, but then we say that peptides and other substances can be used to influence different targets."
are expected to be inserted, in an appropriate form, in the manuscript.
I do not understand why "adding the entry numbers to tables and text can lead to confusion in the structure and confuse the reader". On the contrary, the lack of such linkages impedes understanding of the article.
Additionally, in the new version there is a new figure, apparently in response to the comments by the second Reviewer. I had a similar feeling (however, not included in my review) that some chemical structures should be shown. Unfortunately, the new added Figure 3 is, in my opinion, mainly useless. There are only two informative structures: Micafungin and Polymyxin B. The 7 others have no value in such a general form of 3D peptide structures – there is a countless number of peptides of the same structure. They should be implemented or replaced with their AA sequence. If the authors show 3D models of the peptides, they should also discuss how these structures are important in interactions with the targets. Moreover, apart from providing several structures of AFPs in Fig.3, there should be another figure (or figures) with chemical structures of non-peptide antifungals discussed in the review.
In summary, I am not satisfied with the revised version of the manuscript. It still requires major corrections.
Author Response
Dear Reviewer,
We are grateful to you for the suggestions allowing us the improving of our manuscript.
Please, see our comments to your remarks and the revised text of the paper:
The authors partially improved the manuscript. The answers to my queries seem to be reasonable, but these explanations should be included in the manuscript. It is the readers, not the reviewer. who should know the area of the review. It is the readers, not the reviewer, who should know why some AFPs are discussed in section 2, while some other in subsequent sections.
Thus, the explanations:
"The difference of our review from others is that it contains information not only about antifungal peptides/proteins, but also about other substances that exhibit antifungal activity. In addition, we discuss combining - we lead the reader to the idea of combination and, when choosing objects to present in the review, we were guided by the demonstration of the wide potential of antifungal compounds. Not limited to peptides, but also focused the reader on the possibility of combining these compounds, including peptides, which are very suitable for this because of their properties, enzymes that act on various fungal targets, and with metals that generate reactive oxygen species."
" In our article we considered various antifungals with a wide range of possible application. Since, as we have mentioned in the Introduction, the fungal pathogenesis is a common problem it is necessary to consider antifungals aimed at different targets and area of application."
"As it was mentioned by the Reviewer, today a huge number of peptides are known and covering everything in one article is nearly impossible task. When selecting substances for discussion in our review, we were guided by the number of studies conducted with each of the peptides, the presence of a known structure, as well as the presence of synthetic analogues in the case of the natural peptides considered. The main idea was to show how appropriate it is to obtain certain synthetic analogues of natural peptides and how does these change the peptide properties (antimicrobial efficiency, toxicity, stability, mechanism of action, etc.). At the same time, the most recent and relevant synthetic peptides obtained by prediction and studied against various pathogens were also considered."
"The substances of peptide nature you mentioned are inserted into other tables in accordance with their effect on fungi (target for action). Therefore, transferring them to Table 1 is not consistent with the logic and structure of our review article, which will lead to inconvenience for the reader. We have a slightly original view on the content of the review - to collect not so much substances by their chemical nature, but by the point of impact on the object. At the beginning we write about peptides, since this is a traditional object in research against fungi, but then we say that peptides and other substances can be used to influence different targets."
are expected to be inserted, in an appropriate form, in the manuscript.
Thank you for the recommendations. We have added our relevant explanations to the text of the review.
I do not understand why "adding the entry numbers to tables and text can lead to confusion in the structure and confuse the reader". On the contrary, the lack of such linkages impedes understanding of the article.
We have added the entry numbers to all tables and text of the review.
Additionally, in the new version there is a new figure, apparently in response to the comments by the second Reviewer. I had a similar feeling (however, not included in my review) that some chemical structures should be shown. Unfortunately, the new added Figure 3 is, in my opinion, mainly useless. There are only two informative structures: Micafungin and Polymyxin B. The 7 others have no value in such a general form of 3D peptide structures – there is a countless number of peptides of the same structure. They should be implemented or replaced with their AA sequence. If the authors show 3D models of the peptides, they should also discuss how these structures are important in interactions with the targets. Moreover, apart from providing several structures of AFPs in Fig.3, there should be another figure (or figures) with chemical structures of non-peptide antifungals discussed in the review.
We have updated 3D structures of peptides in Figure 3 and implemented them with amino acid sequences. We also added a brief discussion about the importance of 3D structures and a reference #71 (Masuda, Y. Bioactive 3D structures of naturally occurring peptides and their application in drug design. Biosci. Biotechnol. Biochem. 2021, 85, 24–32. https://doi.org/10.1093/bbb/zbaa008).
Since the text of the review already contains a detailed discussion of how different antifungal substances interact with certain targets to exhibit an antifungal effect (pages 13,15,16,17,20,22), so we did not add same information near the structures of peptides. With all due respect to your suggestion, we don't think it's worth overloading the text with another figure (or figures) with chemical structures of non-peptide antifungals discussed in the review. These are well-known substances and the text contains references to those articles where these substances were described and studied by the authors. We consider that does not reduce the quality of the review.
Thank you again for all your recommendations.
With very kind regards,
The authors of the manuscript
Round 3
Reviewer 2 Report
Comments and Suggestions for Authors
The manuscript is now suitable for publication.
I found 2 typos in l. 131: 'Table1 [27-43]),' should be, I guess, 'Table 1 [27-43],'